# Decentralized Dynamic Cooperation of Personalized Models for Federated Continual Learning

**Danni Yang**[1][*], **Zhikang Chen**[1][*], **Sen Cui**[1][*], **Mengyue Yang**[3], **Ding Li**[1],
**Abudukelimu Wuerkaixi**[1], **Haoxuan Li**[2,6][†], **Jinke Ren**[4], **Mingming Gong**[5,6][†]

[1]Tsinghua University   [2]Peking University   [3]University of Bristol
[4]The Chinese University of Hong Kong, Shenzhen   [5]The University of Melbourne
[6]Mohamed bin Zayed University of Artificial Intelligence (MBZUAI)
{hxli@stu.pku.edu.cn, mingming.gong@unimelb.edu.au}

## Abstract

Federated continual learning (FCL) has garnered increasing attention for its ability to support distributed computation in environments with evolving data distributions. However, the emergence of new tasks introduces both temporal and cross-client shifts, making catastrophic forgetting a critical challenge. Most existing works aggregate knowledge from clients into a global model, which may not enhance client performance since irrelevant knowledge could introduce interference, especially in heterogeneous scenarios. Additionally, directly applying decentralized approaches to FCL suffers from ineffective group formation caused by task changes. To address these challenges, we propose a decentralized dynamic cooperation framework for FCL, where clients establish dynamic cooperative learning coalitions to balance the acquisition of new knowledge and the retention of prior learning, thereby obtaining personalized models. To maximize model performance, each client engages in selective cooperation, dynamically allying with others who offer meaningful performance gains. This results in non-overlapping, variable coalitions at each stage of the task. Moreover, we use coalitional affinity game to simulate coalition relationships between clients. By assessing both client gradient coherence and model similarity, we quantify the client benefits derived from cooperation. We also propose a merge-blocking algorithm and a dynamic cooperative evolution algorithm to achieve cooperative and dynamic equilibrium. Comprehensive experiments demonstrate the superiority of our method compared to various baselines. Code is available at: https://github.com/ydn3229/DCFCL.

## 1 Introduction

Federated learning (FL), as a distributed machine learning framework, addresses privacy and efficiency issues inherent in traditional centralized data processing [1, 2]. Most existing works based on fixed local data distribution aim to minimize a static joint objective. However, in real-world applications, clients continually collect new data over time, which leads to temporal catastrophic forgetting on local sides, a critical challenge in continual learning (CL), which means parameters learned for past tasks drift toward new tasks during training.

To achieve FL in realistic scenarios with dynamic arrival of local data, federated continual learning (FCL) has been proposed. FCL faces two critical challenges: at local training stage, clients need to overcome temporal catastrophic forgetting induced by learning new tasks; at aggregation stage,

---

[*]Equal contribution.
[†]Corresponding authors.

39th Conference on Neural Information Processing Systems (NeurIPS 2025).

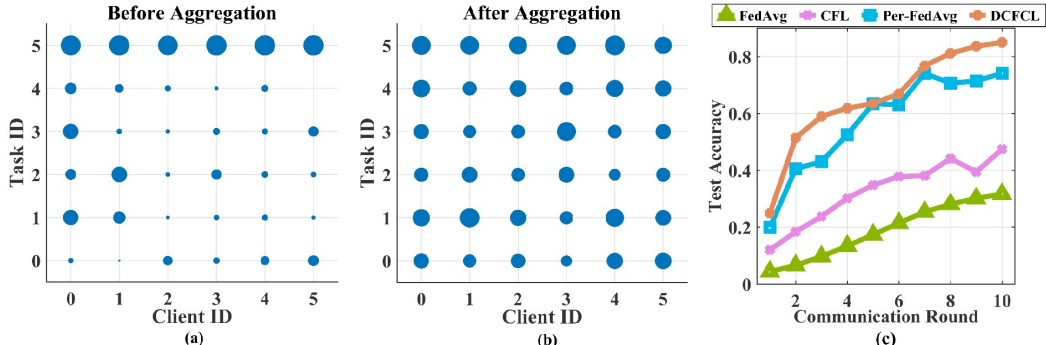

**explain** : (a) and (b) show the effect of model aggregation on catastrophic forgetting. The circle size represents the accuracy of each task on each client, with larger circles indicating higher accuracy and smaller circles indicating lower. (c) shows the impact of decentralized aggregation of personalized models on performance of federated continual learning.

Figure 1: Spatial and temporal catastrophic forgetting in FCL.

spatial catastrophic forgetting should be addressed caused by knowledge interference from aggregated heterogeneous models. However, we assume aggregation can benefit clients in mitigating these issues, as learning from others facilitates acquisition of new knowledge and retention of previous learning. To verify this conjecture, we trial our method on EMNIST [3] with 5 clients, each with 5 tasks in Fig. 1(a)(b), which show test accuracy of before and after aggregation. Before aggregation, models exhibit noticeable drift, heavily favoring new tasks. After that, accuracy on previous tasks improves significantly, underscoring influence of aggregation in alleviating catastrophic forgetting.

Although aggregation can mitigate catastrophic forgetting for personalized models, we believe the effect is uncertain, as clients may have incredible spatial data heterogeneity [4]. Early studies adopt a central server architecture [5, 6] to aggregate, which performs poorly when facing strong heterogeneity. In fact, several decentralized methods have been developed in personalized FL [7, 8]. In Fig. 1(c), we set up heterogeneous scenario on MNIST [9] to illustrate personalized (Per-FedAvg) [10] and decentralized aggregation (CFL) [7] significantly improve performance compared to centralized method (FedAvg) [11]. By group aggregation in decentralization topology, it can promote effectiveness of aggregation and alleviate heterogeneous interference, therefore further mitigates catastrophic forgetting. However, directly transferring decentralization from FL to FCL suffers from ineffective grouping aggregation caused by task changing.

Inspired by above discussion, we introduce a novel decentralized **D**ynamic **C**ooperative **F**ederated **C**ontinual **L**earning (DCFCL) framework to achieve personalized FCL, allowing clients to form non-overlapping coalition topology in each aggregation phase to prevent grouping ineffectiveness. These coalitions are composed by several subsets of clients who assist one another in improving their respective model performance to facilitate personalized learning. We aim to identify coalitions to achieve cooperative equilibrium state, where no alternative coalitions would yield greater benefits for all cooperators inside. Equilibrium is dynamic, capable of disintegration or reorganization as tasks change, eventually leading to new equilibrium.

To achieve above framework, we utilize knowledge distillation to maintain model consistence to identify cooperators, then quantify and calculate client benefits in various coalitions based on overall similarity-comprising gradient coherence and model similarity and coalitional affinity game to further formulate benefit table. After obtaining benefit table, we propose a merge-blocking algorithm to achieve equilibrium state and a dynamic cooperative evolution algorithm to evolve new equilibrium at each aggregation phase. Through dynamic cooperative equilibrium, clients achieve personalized models in decentralized FCL framework. The main contributions of this paper are as follows:

- We propose a novel decentralized framework for personalized FCL, allowing dynamic cooperation among clients to mitigate catastrophic forgetting and improve model performance.
- We use overall similarity and coalitional affinity game to effectively quantify and calculate client benefits in cooperative coalitions.
- We propose merge-blocking algorithm to recognize cooperative equilibrium and dynamic cooperative evolution algorithm to quickly evolve new equilibrium at each aggregation.

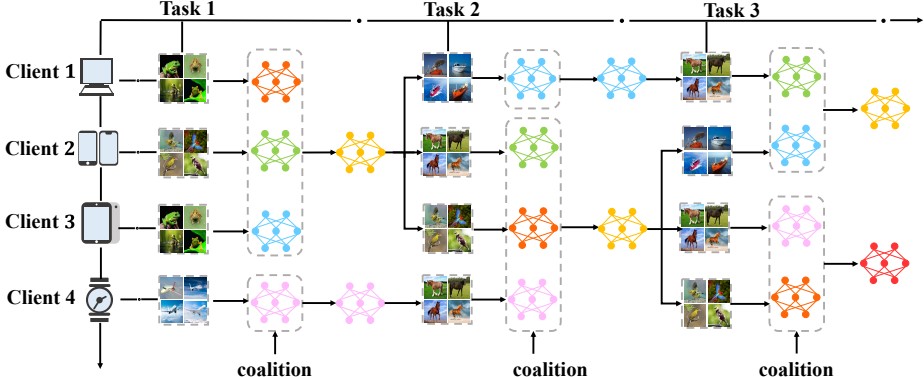

**explain**: In task 1, clients 1, 2, and 3 have similar data distribution, so they cooperate, whereas 4's task differs from them, providing no mutual benefit. Thus, 4 trains locally. In task 2, clients 2 and 4 have similar distributions, leading to cooperate. Meanwhile, 2 cooperates with 3 to recall task 1. The same is true for task 3.

Figure 2: System model. Illustrate dynamic cooperation in decentralized federated continual learning.

## 2 Related Works

**Continual Learning** CL addresses a common scenario in which tasks arrive as continuous data stream for network to learn. Strategies like regularization-based, rehearsal-based, and dynamic architecture-based approaches are employed to mitigate catastrophic forgetting. Regularization-based methods like EWC [12] constrain changes in weights of previous tasks, thereby reducing catastrophic forgetting. Rehearsal-based approaches involve preserving data of previous tasks or generating pseudo-data [13] to train next task, like LUCIR [14] and iCaRL [15]. Dynamic architecture-based methods encompass expanding models or employing parameter isolation to retain previous knowledge, such as Piggyback [16], WSN [17], and LwI [18].

**Federated Learning** FL is typically categorized into centralized and decentralized frameworks. Centralized FL [19] like FedAvg [11], FedProx [20], and SCAFFOLD [21] involve aggregating locally trained models from individual clients on a central server to obtain a global model. Decentralized FL is tailored for client needs. Hypernetworks are introduced to enable decentralized cooperative FL [22, 23]. Decentralized protocol is also proposed to support personalized learning [24, 10].

**Federated Continual Learning** FCL considers not only catastrophic forgetting but also irrelevant knowledge interference. Knowledge distillation is used for knowledge preservation [25, 26, 27]. Replay is also extended from CL to FCL, like FedCIL [28] and AF-FCL [29]. These methods adopting centralization may lead to suboptimal performance once substantial heterogeneity arises.

**Cooperative Game Theory** Cooperative game theory investigates strategy where players can achieve agreements on coalitions and benefits of cooperators [30, 31, 32]. Collaborating in FL is proposed to develop personalized models [23]. Cooperative game is also explored in resolving linear regression and mean estimation problems in FL [33, 34]. These works rely on static cooperative strategy formulating fixed coalitions, which may lose effectiveness due to task variations. So we emphasize dynamic cooperative strategy for FCL.

## 3 Decentralized Federated Continual Learning

### 3.1 Problem Setup

In a decentralized FCL architecture, there are $K$ clients forming the set $\mathcal{K} = \{1, \ldots, K\}$ without a central server. Each client has a local dataset $\mathcal{D}_k = \{\mathcal{D}_k^1, \mathcal{D}_k^2, \ldots, \mathcal{D}_k^T\}$, where $T$ denotes the total number of task phases and $\mathcal{D}_k^t = \{x_k^{ti}, y_k^{ti}\}_{i=1}^{n_k^t}$ is the training data in phase $t$ containing $n_k^t$ samples and $\{x_k^{ti}, y_k^{ti}\}$ is the $i$-th data sample. $y_k^{ti} \in \mathcal{C}_k^t$, and $\mathcal{C}_k^t$ denotes the class set of $\mathcal{D}_k^t$. In practical scenarios, it may be observed that the task set of clients is not necessarily correlated. Thus we consider a practical setting, the limitless task pool (LTP), denoted as $\mathcal{T}$. For each client, the dataset $\mathcal{D}_k^t$ of the $k$-th client at time $t$ corresponds to a particular learning task $\mathcal{T}_k^t \subset \mathcal{T}$. There

is no guaranteed relation among the tasks $\{\mathcal{T}_k^1, \mathcal{T}_k^2, \ldots, \mathcal{T}_k^T\}$ in the $k$-th client at different steps. Similarly, at time $t$, there could be no relation among the tasks $\{\mathcal{T}_1^t, \mathcal{T}_2^t, \ldots, \mathcal{T}_K^t\}$ across different clients, i.e., $\left| \{\mathcal{T}_p^i\}_{i=1}^{t_p} \cap \{\mathcal{T}_q^i\}_{i=1}^{t_q} \right| \geq 0, \ p, q = 1, 2 \ldots K$. More importantly, clients possess diverse joint distributions of data and labels due to heterogeneity. Therefore, at aggregation phase, local models always deviate from their current tasks. Our goal is for decentralized FCL to enable clients acquire new knowledge while retaining prior learning through aggregation. Consequently, at each task phase $t$, model parameter of client $k$ is $\theta_k^t$, and optimization goal of each client is:

$$\underset{\theta_k^t}{\operatorname{argmin}} \mathbb{E}[L_k(\theta_k^t; \mathcal{T}_k^1, \mathcal{T}_k^2, \ldots, \mathcal{T}_k^t)], \tag{1}$$

where $L_k$ is the risk objective of client $k$.

### 3.2 System Model

In decentralized FCL system, dynamic cooperation with others is a good method to enhance the model's performance on current tasks while mitigating catastrophic forgetting of previous tasks. This scenario is illustrated in Fig. 2. Suppose there are four clients, each with three tasks. Because of heterogeneity, the best model for a particular client is likely to come from cooperating with a subset of clients rather than all. At each task stage, clients select different cooperative partners based on the trade-off between acquisition of new knowledge and retention of prior learning. The final cooperation result is an equilibrium state composed of non-overlapping coalitions where all clients are relatively satisfied with their current coalitions and do not shift to other groups. With the constant arriving of new tasks, the equilibrium state for each task phase will evolve dynamically.

Assuming each client has $T$ tasks, during the $\tau$ round of local updates for task $t$. When the coalition structure that client $k$ belongs to is $S$, the aggregated model $\theta_k^\tau$ of client $k$, can be updated by the following steps:

(a) local iterations:

$$\theta_k^{\tau+\frac{1}{2}} \leftarrow \theta_k^\tau - \eta \nabla_\theta L_k^\tau(\theta_k^\tau; \mathcal{D}_k^\tau), \tag{2}$$

followed by aggregation step that updates local model $\theta_k^{\tau+\frac{1}{2}}$ with a combination of model updates $\Delta \theta_k^\tau = \theta_k^{\tau+\frac{1}{2}} - \theta_k^\tau$.

(b) aggregation:

$$\theta_k^{\tau+1} = \alpha_k \theta_k^{\tau+\frac{1}{2}} + \sum_{i \in S \setminus \{k\}} \alpha_i \theta_i^{\tau+\frac{1}{2}} = \alpha_k(\theta_k^\tau + \Delta\theta_k^\tau) + \sum_{i \in S \setminus \{k\}} \alpha_i(\theta_i^\tau + \Delta\theta_i^\tau) = \sum_{i \in S} \alpha_i(\theta_i^\tau + \Delta\theta_i^\tau) \tag{3}$$

where $\alpha_i$ can be explained as weight coefficient of client $i$. Therefore, the optimization variable of 1 is determined by steps (a)(b) simultaneously, which can be subdivided into $\theta_k^{\tau-1}, S | k \in S$.

### 3.3 Cooperative Game

To achieve the optimization goal shown in 1 in the above-mentioned system model, we introduce the concept of cooperative game, which is usually modeled as a process of coalition formation [35]. Using language of cooperative game theory, we can interpret a cooperative state $s_m^\tau$ at round $\tau$ as a partition $\pi$ consisted of non-overlapping coalitions between clients, as well as benefit vector $u(\pi)$ for each client, i.e., $s_m^\tau = (u(\pi), \pi)$. There are $B_K$ states for $K$ clients forming a set $\mathcal{S}^\tau = \{s_1^\tau, \cdots, s_{B_K}^\tau\}$. For any state $s_m^\tau$, $u_k(\pi)$ denotes benefit to $k$ under corresponding partition $\pi$. We aim to find an optimal state that yields $\theta_k^\tau$ minimizing loss while maximizing benefit (i.e. $u_k(s_m^\tau) := -L_k(\theta_k^\tau; D_k^{val})$), which can be achieved by: $u_k(s_*^\tau) = \max_m u_k(s_m^\tau) = \max_{S | S \in \pi(s_m^\tau)} -L_k(\sum_{i \in S} \alpha_i \theta_i^\tau; D_k^{val}) = \max_{\theta_k^\tau} -L_k(\theta_k^\tau; D_k^{val}) = \min_{\theta_k^\tau} L_k(\theta_k^\tau; D_k^{val})$, where $\theta_k^\tau = \sum_{i \in S} \alpha_i(\theta_i^{\tau-1} + \Delta\theta_i^{\tau-1}) = \sum_{i \in S} \alpha_i \theta_i^\tau$. $S \in \pi(s_m^\tau)$ is coalition that client $k$ belongs to. The optimization problem of 1 becomes problem of cooperative game after local iteration and optimization variables include local model parameter $\theta_i^{\tau-1}$ and coalition structure $S$. The coalition set is $\mathbb{S} = \{S_1, \cdots, S_{2^K-1}\}$ including all coalitions for $K$ clients. Based on different coalitions, clients can obtain various benefits. These coalitions and benefits can eventually formulate a benefit table.

**Figure 3** (a) benefit table:

| State | Partition | Benefit | | |
|---|---|---|---|---|
| | | $C_1$ | $C_2$ | $C_3$ |
| $s_1^\tau$ | $\{C_1\},\{C_2\},\{C_3\}$ | 0 | 0 | 0 |
| $s_2^\tau$ | $\{C_1, C_2\},\{C_3\}$ | 1 | 2 | 0 |
| $s_3^\tau$ | $\{C_1, C_3\},\{C_2\}$ | 2 | 0 | 1 |
| $s_4^\tau$ | $\{C_1\},\{C_2, C_3\}$ | 0 | 4 | 2 |
| $s_5^\tau$ | $\{C_1, C_2, C_3\}$ | 2 | 1 | 2 |

(b) Transition Process:

| State | 1 | 2 | 3 | 4 |
|---|---|---|---|---|
| $s_1^\tau$ | | | | |
| $s_2^\tau$ | | | | |
| $s_3^\tau$ | | | | |
| $s_4^\tau$ | | | | |
| $s_5^\tau$ | | | | |

(c) Next State:

| State | 1 | 2 | 3 |
|---|---|---|---|
| $s_1^\tau$ | $s_2^\tau$ | $s_3^\tau$ | $s_4^\tau$ |
| $s_2^\tau$ | $s_3^\tau$ | $s_4^\tau$ | $s_4^\tau$ |
| $s_3^\tau$ | $s_4^\tau$ | $s_4^\tau$ | $s_4^\tau$ |
| $s_4^\tau$ | $s_4^\tau$ | $s_4^\tau$ | $s_4^\tau$ |
| $s_5^\tau$ | $s_4^\tau$ | $s_4^\tau$ | $s_4^\tau$ |

(d) partition changes.

(a)  (b)  (c)  (d)

**explain**: (a) benefit table; (b) is transition process of equilibrium formation. Arrows indicate transitions from previous one to next. Each state is a different color and eventually reaches equilibrium $s_4^\tau$; (c) is next state at each transition corresponding to (b); (d) shows partition changes, and red dotted line represents blocking coalition which contributes to transition from previous to next.

Figure 3: Benefit table and state transition process with three clients as an example.

**Achieving equilibrium for stable cooperation**   Fig. 3(a) shows an example of a benefit table with 3 clients, including 5 cooperative states, 5 partitions and 7 coalitions. Obviously, there is no coalition partition that allows all clients to reach their optimal benefit simultaneously. However, given the limited state space of coalition partitions, there is at least one equilibrium state where all clients are relatively satisfied with benefit in current coalition and will not deviate to other groups. To achieve the equilibrium state, we propose the concept of the transition process of equilibrium formation (TPEF), which involves transitioning from one state to another, ultimately reaching equilibrium. Transitions are driven by clients who can derive better benefits from forming coalition, known as profitable transition (PT). Assuming a state $s_m^\tau$ and a coalition $S$, then $S$ has a weak PT from $s_m^\tau$ if there is a state $s_n^\tau$ with $S \in \pi(s_n^\tau)$ such that $u_k(s_n^\tau) \geqslant u_k(s_m^\tau)$ for all $k \in S$, which means some clients can obtain the same or more benefits by forming coalitions with each other. When $\geq$ turns into $>$, all clients can get more benefits than now, changing to a strict PT. Here $S$ is called blocking coalition $(BC)$. If there is a strict PT, state must transfer. Once there is a client in $S$ suffer from benefit loss, state doesn't change. $s_m^\tau$ is equilibrium state if there is no coalition state $s_n^\tau$ with a blocking coalition $S$ such that $\forall k \in S$, if $k \in S_i, 1 \le i \le m$ then $u_k(s_n^\tau) \geqslant u_k(s_m^\tau)$ and $\exists l \in S$, if $l \in S_j, 1 \le j \le m$, then $u_l(s_n^\tau) > u_l(s_m^\tau)$. As shown in Fig. 3(b)(c), transition process is listed. At $s_1^\tau$ the coalition $\{C_1, C_2\}(BC)$ leads to better benefits for each, thus $C_1, C_2$ will cooperate, state transfers to $s_2^\tau$. At $s_3^\tau$, $C_3$ will betray $\{C_1, C_3\}$ and switch to $\{C_2, C_3\}(BC)$, and state will transfer to $s_4^\tau$. Any state will eventually transfer to $s_4^\tau$, which has no $BC$ for it and thus represents equilibrium.

# 4   Dynamic Cooperative Strategy

Our goal is to develop a dynamic cooperative strategy that achieves equilibrium at each aggregation stage. To accomplish this, we need to complete two key tasks: (1) Formulating benefit table. The most intuitive method involves creating various aggregation models based on different coalitions. These aggregation models are then used to test performance of all tasks on local clients, which can determine client benefits. Theoretically, there are $B_K$ cooperative states for $K$ clients, where $B_K$ is Bell number representing the number of ways to partition a set with $K$ elements. Given that exhaustively trying all aggregation models locally has extremely heavy computation and communication cost, we propose concept of overall similarity among clients to quantify 2-client benefits. Then, we use coalitional affinity game to quickly calculate multi-client benefits. (2) Achieving dynamic cooperative equilibrium. Based on analysis of TPEF in 3.3, traversing TPEF of all states can find equilibrium, however it requires exponential time complexity, so we explore efficient merge-blocking algorithm to achieve equilibrium and dynamic cooperative evolution algorithm to quickly evolve new equilibrium.

## 4.1   Preparatory Condition

**Knowledge distillation for maintaining consistent features to identify cooperator**   When client trains on new task, the classifier is continuously modified by new features, which is not conducive to identifying cooperators who can assist in recalling previous knowledge. Therefore, we maintain the consistency of the classifier's feature space to maximize the utilization of their own model information rather than extra information exchanging to identify cooperators efficiently.

We apply knowledge distillation in classifier to control classifier's feature space preventing from drift to new task. First, there is one teacher model (past model of round $\tau - 1$) and one student model

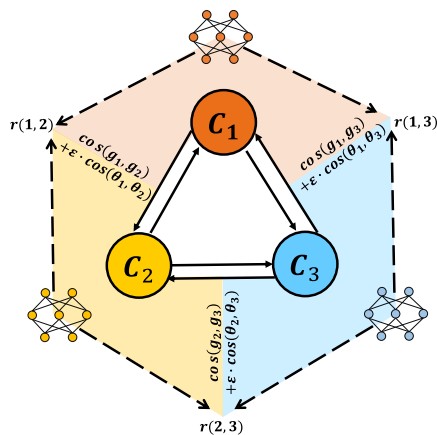

Figure 4: An affinity graph for 3-client coalition.

(current round $\tau$). Output logits for teacher model are denoted as $\mathbf{o}^{\tau-1}(x) = [o_1^{\tau-1}(x), \ldots, o_n^{\tau-1}(x)]$, where $x$ is an input to network and $n$ is the dimension of logits vector, and logits of student model are $\mathbf{o}^\tau(x) = [o_1^\tau(x), \ldots, o_n^\tau(x)]$. The distillation loss for client $k$ on round $\tau$ is defined as:

$$L_{dis}^\tau(\theta_k^\tau; \mathcal{D}_k^\tau) = \sum_{x \in \boldsymbol{x}_k^\tau} \sum_{i=1}^{n} -p_i^{\tau-1}(x) \log \left[ p_i^\tau(x) \right], \tag{4}$$

where $\theta_k^\tau$ is student model, and $p_i^{\tau'}(x) = \frac{e^{o_i^{\tau'}(x)/\mathcal{F}}}{\sum_{j=1}^{n} e^{o_j^{\tau'}(x)/\mathcal{F}}}$ are temperature-scaled logits, where $\mathcal{F}$ is temperature scaling parameter. $p_i^{\tau-1}$ refer to predictions of teacher model ($\mathbf{o}^{\tau-1}(x)$) and $p_i^\tau(x)$ refer to student model ($\mathbf{o}^\tau(x)$). The classification loss in FCL is

$$L_{class}^\tau(\theta_k^\tau; \mathcal{D}_k^\tau) = \sum_{(x,y) \in \mathcal{D}_k^\tau} \sum_{i=1}^{n} -y_i \log \frac{exp(o_i^\tau(x))}{\sum_{j=1}^{n} exp(o_j^\tau(x))}, \tag{5}$$

The final loss can be formulated as

$$L_k^\tau = L_{class}^\tau + \lambda L_{dis}^\tau. \tag{6}$$

where $\lambda$ is a scalar which regularizes influence of $L_{dis}^\tau$.

## 4.2 Formulating Benefit Table

In order to form a complete benefit table, we first propose concept of overall similarity to quantify benefits of 2-client coalition. Taking this as backbone, we calculate benefits of multi-client coalition based on theory of coalitional affinity game.

**Benefit quantification with overall similarity** To reduce communication and computing overhead, we utilize the model information rather than extra information exchanging to quantify client benefits. It is highlighted that finding a descending direction close to the local gradient for aggregating models can reduce conflicts caused by client heterogeneity [36, 37]. Inspired by this, we first quantify benefits through local model gradient coherence. However, relying solely on gradient coherence may aggregate heterogeneous models generating clients interference. This is because the model parameters of different clients may differ significantly overall, even if their gradients are similar. Therefore, we propose to incorporate global model similarity, as it contains essential global information. We comprehensively utilize these two similarity measures as an overall similarity, considering both the coherence of gradient direction and the proximity of model parameters. For ease of representation, at a communication round $\tau$, we use $g_i$, $g_j$ to represent the gradient of client $i$ and $j$, and $\theta_i$ and $\theta_j$ to represent the model parameters. We use cosine similarity to calculate. Therefore, benefits under 2-client coalition can be defined as overall similarity of $i$ and $j$, i.e.,

$$u_i = u_j = \cos(g_i, g_j) + \varepsilon * \cos(\theta_i, \theta_j) = \frac{<g_i, g_j>}{||g_i|| \cdot ||g_j||} + \varepsilon * \frac{<\theta_i, \theta_j>}{||\theta_i|| \cdot ||\theta_j||} = a_{ij} + \varepsilon * b_{ij} \tag{7}$$

where $a_{ij}$ and $b_{ij}$ represent gradient cosine similarity and model cosine similarity of $i$ and $j$, respectively. $\varepsilon$ is a hyperparameter, when it equals to 0, only gradient similarity represents benefits.

**Benefit calculation with coalitional affinity game** With the benefits of 2-client, we need to calculate benefits of multi-client coalition. Coalitional affinity game is a solution because it can model relationships between clients. It is a kind of hedonic game that explicitly models the value that an agent receives from being cooperated with other agents [38]. We can use it to infer benefits in the multi-client coalition through the relationship between two clients. For any pair of clients, we denote affinity of $i$ for $j$ as $r(i, j) \in R$ which represents benefit that $i$ receives from cooperating with $j$, and it is already quantified as overall similarity. We represent the clients and their affinities with an affinity graph $G = \{N, R\}$, it is a weighted directed graph where edge $r(i, j) \in R$ represents an affinity relation between $i$ and $j$. Taking 3-client coalition as an example in Fig. 4, benefits for 2-client are weights on edges in affinity graph. According to affinity graph, benefit of $i$ in multi-client coalition can be defined as the function $f(\cdot)$ of benefit in 2-client coalition, i.e.,

$$
u_i = \begin{cases} 0, & if\ S = \{i\} \\ r(i, j), & if\ S = \{i, j\} \\ f(r(i, j_1), \cdots, r(i, j_n)), & if\ S = \{i, j_1, \cdots j_n\} \end{cases} \tag{8}
$$

Next, we prove the specific format of $f(\cdot)$ in Appendix A. Theoretically, benefit of $i$ is:

$$
u_i = \cos\left(g_{avg}, g_i\right) + \varepsilon * \cos\left(\theta_{avg}, \theta_i\right) = \frac{\sum_{p \in S \setminus \{i\}} \alpha_p a_{ip} ||g_p||}{\sqrt{\sum_{p \in S \setminus \{i\}} \alpha_p^2 ||g_p||^2 + I}} + \frac{\varepsilon \sum_{p \in S \setminus \{i\}} \alpha_p b_{ip} ||\theta_p||}{\sqrt{\sum_{p \in S \setminus \{i\}} \alpha_p^2 ||\theta_p||^2 + H}}
$$

$$
\tag{9}
$$

where

$$
\begin{aligned}
I &= \sum_{p,q \in S \setminus \{i\}, p \neq q} 2\alpha_p \alpha_q g_p g_q = \sum_{p,q \in S \setminus \{i\}, p \neq q} 2\alpha_p \alpha_q a_{pq} \parallel g_p \parallel \parallel g_q \parallel \\
H &= \sum_{p,q \in S \setminus \{i\}, p \neq q} 2\alpha_p \alpha_q \theta_p \theta_q = \sum_{p,q \in S \setminus \{i\}, p \neq q} 2\alpha_p \alpha_q b_{pq} \parallel \theta_p \parallel \parallel \theta_q \parallel
\end{aligned} \tag{10}
$$

and $p \in S \setminus \{i\}$ represents all clients in $S$ except $i$. $\theta_{avg}$ is aggregated model of coalition. To sum up, we define the benefit of $i$ who belongs to coalition $S$ as

$$
u_i = \begin{cases} 0, & if\ S = \{i\}, \\ \cos\left(g_i, g_j\right) + \varepsilon \cos\left(\theta_i, \theta_j\right), & if\ S = \{i, j\}, \\ \cos\left(g_{avg}, g_i\right) + \varepsilon \cos\left(\theta_{avg}, \theta_i\right) & if\ S = \{i, j_1, \cdots j_n\} \end{cases} \tag{11}
$$

We use the form of the weighted average of samples for model aggregation, where

$$
\begin{aligned}
g_{avg} &= \frac{1}{\sum_{p \in S \setminus \{i\}} n_p} \sum_{p \in S \setminus \{i\}} g_p \cdot n_p, \\
\theta_{avg} &= \frac{1}{\sum_{p \in S \setminus \{i\}} n_p} \sum_{p \in S \setminus \{i\}} \theta_p \cdot n_p,
\end{aligned} \tag{12}
$$

where $\alpha_p = \frac{n_p}{\sum_{p \in S \setminus \{i\}} n_p}$. $n_p$ represents sample number of $p$. At task $t$, it equals to $n_p^t$. According to 9, benefit of $i$ in multi-client coalition can be represented by benefit in 2-client coalition. On account of this, we can formulate benefit table quickly by 2-client relationship.

## 4.3 Dynamic Cooperative Equilibrium

Based on analysis of TPEF in 3.3, traversing all states is a method to achieve equilibrium. However, it is computationally intensive, with a time complexity of $O((B_K)^2 K)$. In [39], a merge-split algorithm is used for coalition formation, but it only identifies local optimal solutions in the Pareto Order. Rational clients can benefit more by blocking coalitions in PFCL, therefor equilibrium is ultimately

**Algorithm 1:** Merge-Blocking Algorithm

---

**Input:** The initial partition $\pi_{in}$
**Output:** The final partition $\pi^*$
Sort coalitions set $\mathbb{S}$ in ascending order by the number of clients of each coalition;
Set $\pi_{up} \leftarrow \pi_{in}, \pi_{prev} \leftarrow \emptyset$, Count Table $CT \leftarrow \emptyset$, Stable Coalition $SC \leftarrow \emptyset, \pi^* \leftarrow \emptyset$;
**while** $\pi_{up} \neq \pi_{prev}$ *and* $\pi_{up} \neq \emptyset$ **do**
    Set $\pi_{prev} \leftarrow \pi_{up}$;
    Set $CT \leftarrow \emptyset$;
    **for** $S \in \mathbb{S}$ **do**
        $\pi_{up} = \{S_1, \cdots, S_z\}, \pi_{new} = \{S \cup \pi'_{up}\}$;
        $\pi'_{up}$ is the new set after coalitions in it has removed the elements contained in $S$;
        **if** $all(u_i(\pi_{new}) \geq u_i(\pi_{up})|i \in S)$ *and* $any(u_i(\pi_{new}) > u_i(\pi_{up})|i \in S)$ **then**
            Set $\pi_{up} \leftarrow \pi_{new}$;
            Remove *all* $S_i \in CT$ *with* $S_i \notin \pi_{up}$ and add counts in $CT$ of $S_i \in \pi_{up}$;

    **if** $len(CT) \neq 0$ **then**
        Set $SC \leftarrow max(CT), \pi_{up} \leftarrow \pi_{up} \setminus SC$;
        **for** $S \in \mathbb{S}$ **do**
            **if** $set(S)\&set(SC)$ **then**
                $\mathbb{S} \leftarrow \mathbb{S} \setminus S$;
        $\pi^* \leftarrow \pi^* \cup SC$;
    **else**
        $\pi^* \leftarrow \pi^* \cup \pi_{up}$;

---

stable result. Motivated by this, we develop a merge-blocking algorithm to achieve cooperative equilibrium and iteratively evolve new equilibrium through dynamic cooperative evolution algorithm.

In Algorithm 1, we traverse coalitions, which have less quantity than states, to reduce computation. We begin with singletons for each client as initial partition and iteratively traverse coalition set $\mathbb{S}$. When current partition encounters a $BC$: $S \in \mathbb{S}$, clients in partition are merged forming $S$ and previous coalitions are blocked and reorganized. To reduce traversals, we introduce stable coalition $(SC)$ to prune. By tracking the frequency of coalitions in update partition, we identify $SC$ with maximum counts accumulated and cannot be blocked by any other $BC$s. Then we remove all coalitions from $\mathbb{S}$ which contain clients of $SC$ and then continue traverse $\mathbb{S}$ to find the next $SC$ until there is no $BC$. Ultimately, equilibrium partition is the collocation of all $SC$s. Our simulation results indicate that Algorithm 1 converges to equilibrium within only a few traversals. After achieving equilibrium, we evolve new equilibrium by dynamic cooperative evolution algorithm to realize dynamic equilibrium among clients at each aggregation stage. See Appendix for more details.

## 5 Experiments

### 5.1 Experimental Settings

**Datasets and baselines.** We conduct 4 datasets on different settings. **1) EMNIST-LTP [3]:** a character classification dataset with 26 classes. **2) EMNIST-shuffle [3]:** the task sets of EMNIST are arranged in different orders. **3) CIFAR100 [40]:** a challenging image classification datase. **4) MNIST-SVHN-F [9, 41, 42]:** The dataset is constructed with MNIST [9], SVHN [41] and FashionMNIST [42]. We compare our method with 5 FL baselines, 2 CL baselines and 6 FCL baselines. See Appendix for more details of dataset settings and baselines.

### 5.2 Experimental Results on All Datasets

In EMNIST-LTP dataset, clients may encompass unrelated tasks, thus rendering the dataset challenging. The performance of all methods on EMNIST-LTP is shown in Table 1. Our approach exhibits superior performance across all of the comparative experiments. Different from EMNIST-LTP, EMNIST-shuffle represents a more tractable dataset within the conventional setting, resulting in higher overall accuracy rates as in Table 1. Our method still showcases a superior capacity than all

Table 1: Average accuracy on all datasets.

| Model | EMNIST-LTP | EMNIST-shuffle | CIFAR100 | MNIST-SVHN-F |
|---|---|---|---|---|
| FedAvg | $32.5_{\pm 0.9}$ | $70.3_{\pm 0.4}$ | $26.3_{\pm 2.5}$ | $55.7_{\pm 1.4}$ |
| FedProx | $35.3_{\pm 0.5}$ | $69.4_{\pm 0.9}$ | $28.7_{\pm 1.4}$ | $56.1_{\pm 1.0}$ |
| SCAFFOLD | $35.1_{\pm 0.7}$ | $74.7_{\pm 0.5}$ | $37.4_{\pm 1.2}$ | $41.6_{\pm 0.9}$ |
| CFL | $44.5_{\pm 0.6}$ | $71.6_{\pm 0.3}$ | $35.1_{\pm 1.0}$ | $59.2_{\pm 1.0}$ |
| Per-FedAvg | $46.2_{\pm 1.2}$ | $75.2_{\pm 0.9}$ | $35.9_{\pm 1.9}$ | $54.1_{\pm 1.3}$ |
| PODNet+FedAvg | $36.9_{\pm 1.3}$ | $71.0_{\pm 0.4}$ | $30.5_{\pm 0.8}$ | $54.2_{\pm 0.8}$ |
| PODNet+FedProx | $40.4_{\pm 0.4}$ | $70.6_{\pm 0.7}$ | $32.5_{\pm 0.5}$ | $56.4_{\pm 0.4}$ |
| ACGAN+FedAvg | $38.4_{\pm 0.2}$ | $70.0_{\pm 0.5}$ | $32.1_{\pm 1.6}$ | $56.0_{\pm 0.7}$ |
| ACGAN+FedProx | $41.3_{\pm 0.9}$ | $70.3_{\pm 1.2}$ | $31.8_{\pm 0.7}$ | $56.4_{\pm 2.1}$ |
| FLwF2T | $40.1_{\pm 0.3}$ | $71.0_{\pm 0.9}$ | $30.2_{\pm 0.7}$ | $54.2_{\pm 0.6}$ |
| FedCIL | $42.0_{\pm 0.6}$ | $71.1_{\pm 0.4}$ | $33.5_{\pm 0.7}$ | $57.2_{\pm 1.7}$ |
| GLFC | $40.1_{\pm 0.8}$ | $74.9_{\pm 0.6}$ | $35.6_{\pm 0.6}$ | $61.8_{\pm 0.8}$ |
| AF-FCL | $47.5_{\pm 0.3}$ | $75.8_{\pm 0.7}$ | $36.3_{\pm 0.3}$ | $\mathbf{68.1}_{\pm 0.7}$ |
| AFCL | $45.6_{\pm 0.7}$ | $77.0_{\pm 0.6}$ | $32.3_{\pm 0.7}$ | $62.4_{\pm 0.6}$ |
| FPPL | $41.4_{\pm 0.6}$ | $76.1_{\pm 0.9}$ | $31.5_{\pm 0.6}$ | $61.7_{\pm 0.9}$ |
| DCFCL | $\mathbf{52.5}_{\pm 0.7}$ | $\mathbf{78.3}_{\pm 0.6}$ | $\mathbf{40.4}_{\pm 0.8}$ | $66.7_{\pm 0.9}$ |

baselines in this commonly adopted dataset setting. In addition, as data heterogeneity becomes more severe (from EMNIST-shuffle to EMNIST-LTP), our method achieves greater performance compared to others. This is likely because increased data heterogeneity leads to substantial variations among models. Consequently, aggregating knowledge from clients into a global model potentially result in conflicting knowledge. In such scenarios, our decentralized federated learning is more effective.

Table 1 also displays the results of two more challenging datasets: CIFAR100 and MNIST-SVHN-F. By aggregating highly correlated models, our method guarantees client benefits in terms of both optimization direction and global consistency, significantly exceeding performance of most baselines.

## 5.3 Ablation Studies

Our method consists of three main components: (i) Cooperative Equilibrium (CE). We introduce Dynamic Cooperation in decentralized FCL. Global cooperation transfers to **FedAvg**, and non-cooperation degenerates into **Local** algorithm, where clients execute the CL process locally without any aggregation. (ii) Knowledge Distillation (KD). We use knowledge distillation loss to maintain consistent features of classifier during training to identity cooperator, as it can prevent feature drifts. (iii) Overall Similarity (OS). To quantify client benefits, we propose overall similarity. When $\varepsilon$ approaches 0, it degrades to only use gradient coherence for quantification.

Table 2: Ablation studies on EMNIST-LTP and EMNIST-shuffle datasets.

| Model | EMNIST-LTP | EMNIST-shuffle |
|---|---|---|
| w/o CE-FedAvg | $32.5_{\pm 0.9}$ | $70.3_{\pm 0.4}$ |
| w/o CE-Local | $12.3_{\pm 0.6}$ | $17.3_{\pm 0.9}$ |
| w/o KD | $50.3_{\pm 0.3}$ | $73.2_{\pm 0.4}$ |
| w/o OS | $45.3_{\pm 0.8}$ | $73.7_{\pm 0.3}$ |
| DCFCL | $\mathbf{52.5}_{\pm 0.7}$ | $\mathbf{78.3}_{\pm 0.6}$ |

We conduct ablation studies on EMNIST-LTP and EMNIST-shuffle datasets as displayed in Table 2. Our method achieves optimal performance with all three modules. The accuracy of **Local** is incredibly low, which reflects the significance of decentralized cooperation for FCL.

## 5.4 Results for Different Parameter Settings

We conduct experiments on EMNIST-LTP and EMNIST-shuffle datasets with various $\lambda$ and $\varepsilon$. $\varepsilon$ is fixed at 0.2 when $\lambda$ is varied, and vice versa. As shown in Table 3, emphasizing model similarity by increasing $\varepsilon$ enables clients to identify peers with more aligned feature spaces for learning. Therefore, it is essential to determine the optimal overall similarity composition. In addition, we also adjust $\lambda$ to illustrate the influence of knowledge distillation. Increasing $\lambda$ retains more prior task information

for cooperator identification, which in turn promotes more effective cooperation and alleviates catastrophic forgetting.

Table 3: Average accuracy on EMNIST-LTP and EMNIST-shuffle datasets with variable parameters.

| Parameter | | EMNIST-LTP | EMNIST-shuffle | Parameter | | EMNIST-LTP | EMNIST-shuffle |
|---|---|---|---|---|---|---|---|
| $\varepsilon$ | 0.0 | $44.3_{\pm 0.8}$ | $75.9_{\pm 0.3}$ | $\lambda$ | 0.0 | $50.3_{\pm 0.3}$ | $73.2_{\pm 0.4}$ |
| | 0.2 | $\mathbf{52.5}_{\pm 0.7}$ | $78.3_{\pm 0.6}$ | | 0.2 | $52.5_{\pm 0.7}$ | $78.3_{\pm 0.6}$ |
| | 0.4 | $48.7_{\pm 0.7}$ | $\mathbf{80.2}_{\pm 0.6}$ | | 0.4 | $50.7_{\pm 0.2}$ | $78.7_{\pm 0.4}$ |
| | 0.6 | $45.1_{\pm 1.0}$ | $70.4_{\pm 0.5}$ | | 0.6 | $51.3_{\pm 0.7}$ | $74.0_{\pm 0.6}$ |
| | 0.8 | $46.5_{\pm 0.3}$ | $71.4_{\pm 0.6}$ | | 0.8 | $53.7_{\pm 0.8}$ | $77.7_{\pm 0.4}$ |
| | 1.0 | $47.1_{\pm 0.5}$ | $72.2_{\pm 0.8}$ | | 1.0 | $\mathbf{55.7}_{\pm 0.6}$ | $\mathbf{81.3}_{\pm 0.2}$ |

# 6 Conclusion

This study pays attention to critical challenges of temporal and spatial catastrophic forgetting in federated continual learning. We propose a decentralized dynamic cooperative learning framework that personalizes client models. Clients form non-overlapping dynamic coalitions at each aggregation stage to mitigate catastrophic forgetting and further improve performance. The experimental results clearly demonstrate its effectiveness. Whereas, some parameter sensitivity remains(e.g., $\lambda$, $\varepsilon$), which could affect performance in unseen settings. Exploring adaptive mechanisms is left for future work.

## Acknowledgments and Disclosure of Funding

MG was supported by ARC DP240102088 and WIS-MBZUAI 142571. Sen Cui would like to acknowledge the financial support received from Shuimu Tsinghua scholar program. HL was supported by National Natural Science Foundation of China (623B2002). JR was supported by the National Natural Science Foundation of China under Grant No. 62501514.

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

# A   Proof of Theorem

We have obtained the client benefit $r(i, j)$ under overall similarity. If a coalition structure $S$ has 3 client models $x, y, z$, then we have

$$
\begin{aligned}
r(x, z) &= a_{xz} + \varepsilon * b_{xz} \\
r(y, z) &= a_{yz} + \varepsilon * b_{yz} \\
r(x, y) &= a_{xy} + \varepsilon * b_{xy}
\end{aligned}
\tag{13}
$$

where $a_{xy}$ and $b_{xy}$ present gradient coherence and model similarity of $x$ and $y$, respectively. $a_{xz}$ and $b_{xz}$ present gradient coherence and model similarity of $x$ and $z$, respectively. $a_{yz}$ and $b_{yz}$ present gradient coherence and model similarity of $y$ and $z$, respectively.

Then the client benefit of $z$ in $S$ can be defined as the overall similarity of the model to $z$ after $x$ and $y$ are aggregated. The gradient and model after aggregation are respectively

$$
\begin{aligned}
\theta_{avg} &= \alpha_x \theta_x + \alpha_y \theta_y, \\
g_{avg} &= \alpha_x g_x + \alpha_y g_y,
\end{aligned}
\tag{14}
$$

where $\alpha_x$ and $\alpha_y$ can be explained as the aggregation weight of client $x$ and $y$.

For the aggregation model, we have

$$
\begin{aligned}
\theta_{avg}\theta_z &= \alpha_x \theta_x \theta_z + \alpha_y \theta_y \theta_z \\
g_{avg}g_z &= \alpha_x g_x g_z + \alpha_y g_y g_z \\
\|\theta_{avg}\| &= \sqrt{\alpha_x^2 \|\theta_x\|^2 + \alpha_y^2 \|\theta_y\|^2 + 2\alpha_x \alpha_y \theta_x \theta_y} \\
\|g_{avg}\| &= \sqrt{\alpha_x^2 \|g_x\|^2 + \alpha_y^2 \|g_y\|^2 + 2\alpha_x \alpha_y g_x g_y}.
\end{aligned}
\tag{15}
$$

Then the client benefit of $z$ can be expressed as

$$
\begin{aligned}
u_z &= \cos(g_{avg}, g_z) + \varepsilon * \cos(\theta_{avg}, \theta_z) \\
&= \frac{\alpha_x g_x g_z + \alpha_y g_y g_z}{\|g_{avg}\| \cdot \|g_z\|} + \varepsilon * \frac{\alpha_x \theta_x \theta_z + \alpha_y \theta_y \theta_z}{\|\theta_{avg}\| \cdot \|\theta_z\|} \\
&= \frac{\alpha_x a_{xz}\|g_x\| + \alpha_y a_{yz}\|g_y\|}{\sqrt{\alpha_x^2 \|g_x\|^2 + \alpha_y^2 \|g_y\|^2 + 2\alpha_x \alpha_y a_{xy}\|g_x\| \cdot \|g_y\|}} \\
&\quad + \varepsilon * \frac{\alpha_x b_{xz}\|\theta_x\| + \alpha_y b_{yz}\|\theta_y\|}{\sqrt{\alpha_x^2 \|\theta_x\|^2 + \alpha_y^2 \|\theta_y\|^2 + 2\alpha_x \alpha_y b_{xy}\|\theta_x\| \cdot \|\theta_y\|}}
\end{aligned}
\tag{16}
$$

where

$$
\begin{aligned}
g_x g_z &= a_{xz}\|g_x\| \cdot \|g_z\| \\
\theta_x \theta_z &= b_{xz}\|\theta_x\| \cdot \|\theta_z\| \\
g_y g_z &= a_{yz}\|g_y\| \cdot \|g_z\| \\
\theta_y \theta_z &= b_{yz}\|\theta_y\| \cdot \|\theta_z\|
\end{aligned}
\tag{17}
$$

Similarly, when under the multi-client coalition, assumes that the coalition $S = \{1, 2, \cdots, i, \cdots, n-1\}$, $S \in \pi(s_m^\tau)$. The 2-client benefits between clients are

$$
\begin{aligned}
r(i, 1) &= a_{i1} + \varepsilon * b_{i1} \\
r(i, 2) &= a_{i2} + \varepsilon * b_{i2} \\
r(i, n-1) &= a_{in-1} + \varepsilon * b_{in-1}
\end{aligned}
\tag{18}
$$

**Algorithm 2:** Dynamic Cooperative Evolution Algorithm

---

**Input:** $K$ clients in set $\mathcal{K}$, communication round $\tau$, benefit table with all states $s_m^\tau$, benefit vector $u \leftarrow 0$, initial partition $\pi_{in} \leftarrow \{\{1\}, \cdots, \{K\}\}$, coalitions set $\mathbb{S}$

**Output:** cooperative equilibrium $s_*^\tau$

**for** $p \in \mathcal{K}$ **do**

    Calculate $||g_p||, ||\theta_p||$ of $p$;

    **for** $q = p + 1$ **do**

        Calculate $r(p, q)$;

**for** $S \in \mathbb{S}$ **do**

    **for** $k \in S$ **do**

        Calculate benefit of client $k$ in coalition $S$ based on 11;

**if** $\tau = 0$ **then**

    Set $\pi_{in} \leftarrow \{\{1\}, \{2\}, \cdots, \{K\}\}$;

    Perform Algorithm 1 to get $\pi^*$;

**else**

    Set $\pi_{in} \leftarrow \pi^*$;

    Update benefit table;

    Perform Algorithm 1 to get $\pi^*$;

Set $s_*^\tau \leftarrow (\pi^*, u(\pi^*))$;

---

Then for a client $i$ in $S$, the benefit can be expressed as the overall similarity between the aggregated model of the other models in $S$ excluding $i$ and the model of $i$.

$$
\begin{aligned}
u_i(s_m^\tau) &= \cos(g_{avg}, g_i) + \varepsilon * \cos(\theta_{avg}, \theta_i) \\
&= \frac{\alpha_1 a_{i1}||g_1|| + \cdots + \alpha_{n-1} a_{in-1}||g_{n-1}||}{\sqrt{\alpha_1^2||g_1||^2 + \cdots + \alpha_{n-1}^2||g_{n-1}||^2 + I}} \\
&\quad + \varepsilon * \frac{\alpha_1 b_{i1}||\theta_1|| + \cdots + \alpha_{n-1} b_{in-1}||\theta_{n-1}||}{\sqrt{\alpha_1^2||\theta_1||^2 + \cdots + \alpha_{n-1}^2||\theta_{n-1}||^2 + H}} \\
&= \frac{\sum_{p \in S \setminus \{i\}} \alpha_p a_{ip}||g_p||}{\sqrt{\sum_{p \in S \setminus \{i\}} \alpha_p^2||g_p||^2 + I}} \\
&\quad + \varepsilon \frac{\sum_{p \in S \setminus \{i\}} \alpha_p b_{ip}||\theta_p||}{\sqrt{\sum_{p \in S \setminus \{i\}} \alpha_p^2||\theta_p||^2 + H}}
\end{aligned}
\tag{19}
$$

where $H$ and $I$ are defined in Eq.(10).

By mathematical induction, we can get for client $i$ in the coalition $S = \{1, 2, \cdots, i, \cdots, n\}$. With $r(i, n) = a_{in} + \varepsilon * b_{in}$, we have

$$
\begin{aligned}
u_i(s_m^\tau) &= \cos(g_{avg}, g_i) + \varepsilon * \cos(\theta_{avg}, \theta_i) \\
&= \frac{\alpha_1 a_{i1}||g_1|| + \cdots + \alpha_n a_{in}||g_n||}{\sqrt{\alpha_1^2||g_1||^2 + \cdots + \alpha_n^2||g_n||^2 + I}} \\
&\quad + \varepsilon * \frac{\alpha_1 b_{i1}||\theta_1|| + \cdots + \alpha_n b_{in}||\theta_n||}{\sqrt{\alpha_1^2||\theta_1||^2 + \cdots + \alpha_n^2||\theta_n||^2 + H}} \\
&= \frac{\sum_{p \in S \setminus \{i\}} \alpha_p a_{ip}||g_p||}{\sqrt{\sum_{p \in S \setminus \{i\}} \alpha_p^2||g_p||^2 + I}} \\
&\quad + \varepsilon * \frac{\sum_{p \in S \setminus \{i\}} \alpha_p b_{ip}||\theta_p||}{\sqrt{\sum_{p \in S \setminus \{i\}} \alpha_p^2||\theta_p||^2 + H}}
\end{aligned}
\tag{20}
$$

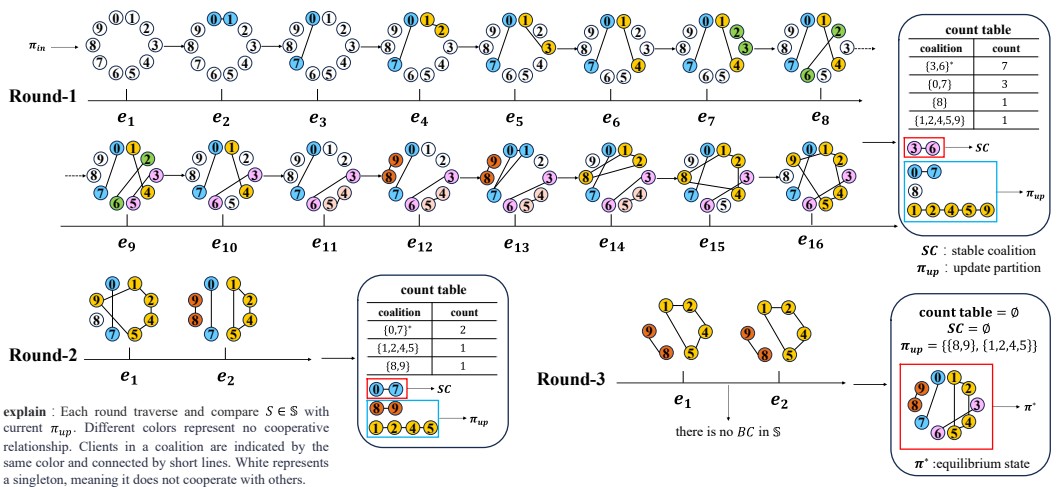

Figure 5: Equilibrium forming process of 10 clients based on Merge-Blocking Algorithm.

# B  Detailed Description of the Algorithm

## B.1  Dynamic Cooperative Evolution Algorithm

With the dynamic arrival of tasks, the equilibrium state is also dynamic following a Markov process, which means the next equilibrium depends solely on the previous equilibrium. We use the dynamic cooperative evolution algorithm to evolve the new equilibrium at each aggregation phase shown in Algorithm 2.

## B.2  Illustrate the Merge-Blocking Algorithm with an Example

We offer 10 clients as example to further illustrate the process of achieving equilibrium in Fig. 5 according to the Algorithm 1 on EMNIST-LTP settings. Initially, all client subsets are generated as the coalition set $\mathbb{S} = [\{0\}, \dots, \{9\}, \{0,1\}, \dots \{0,1,\dots,9\}]$, and the initial partition is $\pi_{in} = [\{0\}, \{1\}, \{2\}, \{3\}, \{4\}, \{5\}, \{6\}, \{7\}, \{8\}, \{9\}], \pi_{up} = \pi_{in}$. At the beginning of the first while loop (Round 1), when comparing with coalition $\{0,1\} \in \mathbb{S}$, the profitable transition (PT) condition is met (i.e., $all\,(u_i\,(\{0,1\}) \geq u_i\,(\pi_{up})|\, i \in \{0,1\}) = 1\ and\ any(u_i(\{0,1\}) \geq u_i(\pi_{up})|i \in \{0,1\}) = 1)$, so the original two coalitions $\{0\}$ and $\{1\}$ in the partition are merged into the blocking coalition ($BC$) $S = \{0,1\}$, and other coalitions remain unchanged, forming new $\pi_{up} = [\{0,1\}, \{2\}, \{3\}, \{4\}, \{5\}, \{6\}, \{7\}, \{8\}, \{9\}]$ at time $e_2$. At $e_{14}$, when compares with $S = \{1,2,8\}$, the original $\{0,1,7\}$ conforms the PT condition, so extracts 1 to cooperate with 2,8 forming $\{1,2,8\}(BC)$ and leaves $\{0,7\}$ to form new $\pi_{up} = [\{1,2,8\}, \{0,7\}, \{3,6\}, \{4,5\}, \{9\}]$. Then continue to traverse $S \in \mathbb{S}$ to compare. After each update, the count of coalitions is accumulated. If $\pi_{up}$ update, the count of changed coalition becomes 0. After traversing $\mathbb{S}$ once, the coalition with the largest count is the stable coalition $SC$, as no $BC$ for it appears. Therefore $\mathbb{S}$ is pruned by removing all coalitions containing clients which belong to $SC$. In next Rounds, $\pi_{up}$ begins to traverse $S \in \mathbb{S}$ again until there is no $BC$ to update the partition, then $\pi_{up}$ is combined with all previous $SC$s to obtain final equilibrium $\pi^*$.

## B.3  Illustrate Dynamic Cooperative Evolution Results on EMNIST-LTP

As shown in Fig. 6, we list equilibrium states at the end round of each task phase on EMNIST-LTP dataset, and it can be seen that the coalition structure changes as the task changes, with clients of the same color forming a coalition. For example, at $t_1$ there are 4 coalitions and 2 coalitions for $t_2$. With the dynamic task flows, cooperative learning through dynamic coalition is necessary. Meanwhile, as the amount of tasks increases, clients tend to form grand coalition to acquire each other's information in order to recall the previous knowledge.

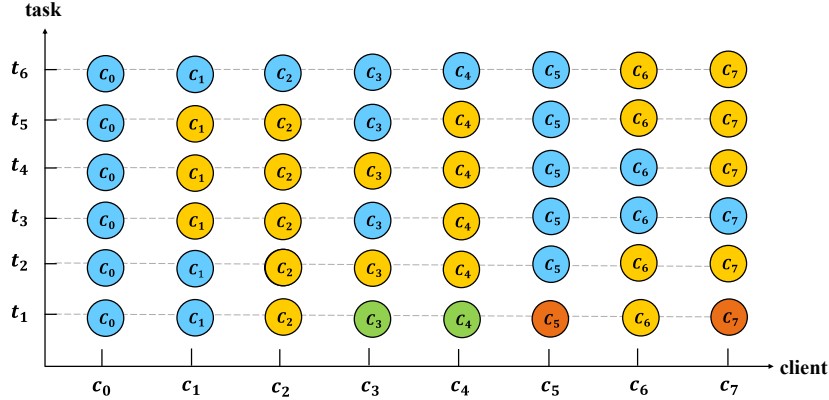

Figure 6: Dynamic Cooperative Evolution on EMNIST-LTP Dataset ($\varepsilon = 0.8, \lambda = 0.2$).

## B.4 Time Complexity Analysis

Suppose that the number of clients is $K$, the number of cooperative states in the FCL system is $B_K$ and the number of coalitions is $2^K - 1$. The analysis of the time complexity is as follows:

(1) Formulating benefit table: In the initialization phase, we only need to measure the overall similarity of 2-client structure, so the complexity of benefit calculation is $O(K^2)$. The complexity of calculating the size of $K$ clients' models is $O(K)$. The values obtained from the initialization can be directly calculated to form the benefit table. Go through all coalitions, each coalition has $k$ clients, and total iteration is $\sum_{k=1}^{K} k * C_K^k = K2^K$, complexity is $O(K2^K)$. Since the complexity of calculating the benefits of the multi-client structure according to the derivation formula is $O(1)$, so the total complexity is $O(K2^K)$. The greatest advantage of our benefit calculation method over other algorithms lies in the fact that we can calculate the individual benefits of rational clients under different groups, rather than only the collective benefits. Additionally, if we aggregate models to calculate the test accuracy on the local client as benefits, the total time complexity is $O(K2^K N)$ if there are $N$ test samples. In contrast, using coalitional affinity game and overall similarity greatly reduces the complexity of formulating benefit table.

(2) Achieving dynamic cooperative equilibrium: Each iteration of merge-blocking algorithm needs to traverse all coalitions and compare the benefits of clients, therefore the complexity is also $O(K2^K)$. The amount of computation is greatly reduced compared to the complexity $O((B_K)^2 K)$ of traversing TPEF of all states in 3.3.

We also list some algorithms using group aggregation for Federated Learning in Table 4 and select representative metrics for contrast.

Table 4: Compare the complexity of different group aggregation algorithms.

| Algorithm | Benefit Calculation [3] | Group Formation | Group's Number | Rational Optimal Solution | Dynamic Group |
|---|---|---|---|---|---|
| ClusterFL | $O(K^2)$ | $O(K^4)$ | 2 | × | × |
| FedGroup | $O(KM)$ | $O(KM^2 + TK^2M)$ | $M$ | × | × |
| Coalitional FL | $O(K^2)$ | $O(\max(K^3, K2^{l_{\max}}))$ | unlimited | × | × |
| pFedSV | - | $O(k!KN)$ | $K$ | × | × |
| DCFCL | $O(K^2)$ | $O(K2^K)$ | unlimited | √ | √ |

ClusterFL quantifies benefits through pairwise similarity and employs Optimal Bipartition Algorithm to minimize inter-group similarity [7]; FedGroup decomposes all weights via Singular Value Decomposition (SVD) into $M$ vectors and applies K-means++ clustering over $T$ iterations [43]; Coalition FL utilizes EMD-based linear combinations of data distributions with Accelerated Device Coalition Formation Algorithm (whose complexity matches ours when $l_{max} = K$, the maximum number of clients) [39]; pFedSV forms coalitions for each client via top-$k$ Shapley values at $O(k!KN)$

---

[3]For fairness, here we only list the benefits calculation under the 2-client structure, as other algorithms do not calculate benefits of multi-clients.

complexity for $N$ test samples [8]. Our algorithm shows following advantages: (1) provides game-theoretically optimal solution for rational clients, though with increased complexity compared to clustering ones [7, 43]; (2) dynamic, scale-unlimited coalition better adapts to continual learning, as evidenced by superior performance; (3) while maintaining computational efficiency through coalition affinity game and structured assumptions (additive/symmetric benefits), we quantify benefits for each client - a feature shared only with Coalition FL - providing a reliable idea for incentives and benefit allocation. Our method's core objective of finding optimal client cooperation in federated continual learning inherently involves computational complexity, as the problem is NP-hard by nature. While our method achieves optimal performance at relatively small scale, practical deployment for large scale necessitates approximating solution, which sacrifices theoretical optimality for computational feasibility, thereby becoming a performance-cost tradeoff.

### B.5 Boarder Impact

To achieve cooperation, all clients must share their model information. This process is facilitated by an impartial and authoritative third party, such as the industry association. The designated third party collects the client models after each round, then assesses the benefits in each state by comparing the overall similarity to determine equilibrium. The cooperative strategies are then published. Therefore, our framework promotes transparent and incentive-aligned cooperation among clients. At the same time, our framework can quantify benefit from each client in a coalition. In practice, such information can be utilized to either provide incentives or to impose charges on each client, to facilitate and enhance the foundation of the coalition.

## C  Implementation Details

### C.1  Datasets

We construct a series of datasets comprising multiple federated clients, with each client possessing a sequence of tasks. Suppose we use $K$ to denote the number of clients, $T$ to denote the number of tasks in each client, and $C$ to denote the number of classes in each task. We curate tasks by randomly selecting several classes from the datasets and sample part of the instances from these classes. Adhering to the principle of class incremental learning, there are no overlapped classes between any two tasks within a client.

**EMNIST-LTP [3].** The EMNIST dataset is a character classification dataset with 26 classes. It contains 145600 instances of 26 English letters. The data contains upper and lower cases with the same label, making classification more challenging. To curate a dataset under LTP setting, we randomly sampled classes from the entire dataset for each client. The EMNIST-LTP dataset consists of 8 clients, with each client encompassing 6 tasks, each task comprising 2 classes ($K = 8, T = 6, C = 2$).

**EMNIST-shuffle [3].** In a conventional reshuffling setting, the task sets are consistent across all clients, while arranged in different orders. Therefore, with the same structure as EMNIST-LTP, we construct EMNIST-shuffle dataset with 8 clients, 6 tasks, and each task comprising 2 classes. While the 6 tasks of all clients are the same but in shuffled orders ($K = 8, T = 6, C = 2$).

**CIFAR100 [40].** As a challenging image classification dataset, CIFAR100 consists of low resolution images containing various objects and complex image backgrounds. We randomly sample 20 classes among 100 classes of CIFAR100 as a task for each of the 10 clients, and there are 4 tasks for each client. For each class, we randomly sample 400 instances into the client dataset ($K = 10, T = 4, C = 20$).

**MNIST-SVHN-F [9, 41, 42].** The dataset is constructed with MNIST [9], SVHN [41] and FashionMNIST [42]. Similar to MNIST, SVHN dataset serves as a benchmark for digit classification tasks, notable for its representation of real-world scenarios with complex backgrounds. We unify the labels of these two datasets. FashionMNIST dataset is designed for clothing image classification. We set 10 clients in the mixed dataset, with each client containing 6 tasks, and each task has 3 classes. In this mixed dataset, different tasks rely on different features. For example, shape features that are relevant to digit classification differ significantly from those that are important for classifying clothing items. Under centralized methods, it may result in incredible knowledge interference ($K = 10, T = 6, C = 3$).

## C.2 Baselines

We compare our method with five baselines from FL, two baselines from CL, and six baselines from FCL. FL methods include basic centralized technique FedAvg, FedProx and SCAFFOLD for reducing heterogeneity interference, decentralized technique CFL for group aggregation, and personalized federated learning method Per-FedAvg. To control variables during local training, we incorporate knowledge distillation into all FL baselines. CL methods are respectively combined with the FL methods (FedAvg, FedProx), training a global model while fighting catastrophic forgetting. The FCL methods focus on addressing the issues of catastrophic forgetting along with statistical heterogeneity.

**Local**. A typical FL comparison method to achieve local training, without global aggregation. In order to control the experimental comparison, we add knowledge distillation to the local training.

**FedAvg** [11]. As a representative FL method, FedAvg trains the models in each client with local dataset and averages their parameters to attain a global model.

**FedProx** [20]. The algorithm is similar to FedAvg. While training local models, a regularization term is employed to govern the proximity between the local parameters and the global parameters. This regularization term serves to effectively control the degree of deviation exhibited by the local models from the global model during the training process.

**SCAFFOLD** [21]. It addresses the issue of client drift by introducing control variates that help align local updates more closely with the global model. This reduces the divergence caused by non-IID data across clients, leading to faster and more stable convergence.

**CFL** [7]. It is designed to optimize federated learning in environments with diverse client data distributions. CFL clusters clients into groups based on their data similarity and trains separate models for each group, allowing for personalized and accurate models while preserving privacy.

**Per-FedAvg** [10]. It is an extension of FedAvg designed to enhance personalization in federated learning. Per-FedAvg focuses on producing a personalized model for each client by incorporating local fine-tuning. This approach balances the benefits of collaborative learning with each client's unique data characteristics.

**PODNet** [44]. A CL method, it incorporates a spatial-based distillation loss onto the feature maps of the classifier. This loss term serves to encourage the local models to align their respective feature maps with those of the previous model, thereby maintaining the performance in previous tasks.

**ACGAN-Replay** [45]. This CL algorithm employs a GAN-based generative replay method. The algorithm trains an ACGAN in the data space to memorize the distribution of previous tasks. While learning new tasks, the classifier is trained on new task data along with generated data from ACGAN.

**FLwF2T** [26]. As a FCL algorithm, FLwF2T leverages the concept of knowledge distillation within the framework of federated learning. It employs both the old classifier from the previous task and the global classifier from the server to train the local classifier.

**FedCIL** [28]. The FCL algorithm extends the ACGAN-Replay method within the federated scenario, addressing the statistical heterogeneity issue with distillation loss.

**GLFC** [27]. In the FCL scenario, the algorithm exploits a distillation-based method to alleviate the issue of catastrophic forgetting from both local and global perspectives.

**AF-FCL** [29] proposes an adaptive forgetting mechanism that dynamically adjusts knowledge retention policies to address catastrophic forgetting in heterogeneous federated learning scenarios.

**AFCL** [6] introduces an asynchronous training paradigm with adaptive synchronization to enable efficient continual learning across heterogeneous federated devices while mitigating forgetting.

**FPPL** [46] introduces a novel federated prototype learning framework that simultaneously addresses catastrophic forgetting and data heterogeneity through efficient prototype propagation and local consistency regularization.

## C.3 Metrics

We use the metrics of accuracy and average forgetting for evaluation works [5, 29]. Suppose $a_k^{i,t}$ is the test set accuracy of the $t$-th task after learning the $i$-th task in client $k$.

Table 5: Average accuracy on CIFAR100 when $K = 8, T = 6, C = 10$.

| Model | CIFAR100 |
|---|---|
| FedAvg | $19.5_{\pm 0.3}$ |
| FedProx | $20.1_{\pm 0.2}$ |
| SCAFFOLD | $20.3_{\pm 0.9}$ |
| CFL | $20.5_{\pm 0.5}$ |
| Per-FedAvg | $29.6_{\pm 1.4}$ |
| PODNet+FedAvg | $21.3_{\pm 0.1}$ |
| PODNet+FedProx | $21.6_{\pm 0.4}$ |
| ACGAN+FedAvg | $19.5_{\pm 0.6}$ |
| ACGAN+FedProx | $19.6_{\pm 0.2}$ |
| FLwF2T | $21.5_{\pm 0.7}$ |
| FedCIL | $19.6_{\pm 0.3}$ |
| GLFC | $19.9_{\pm 0.4}$ |
| DCFCL | $\mathbf{31.4}_{\pm 0.8}$ |

**Average Accuracy**. We evaluate the performance of the model on all tasks in all clients after it finish learning all tasks. By using a weighted average, we calculated the test set accuracy for all seen tasks across all clients, with the number of samples in each task serving as the weights:

$$\text{Average Accuracy} = \frac{1}{\sum_{k=1}^{K} \sum_{t=1}^{T} n_k^t} \sum_{k=1}^{K} \sum_{t=1}^{T} a_k^{T,t} * n_k^t. \tag{21}$$

This approach allows us to account for variations in task difficulty and ensure a fair evaluation across different tasks and clients.

**Average Forgetting**. The metric of average forgetting assesses the extend of backward transfer during continual learning, quantified as the disparity between the peak accuracy and the ending accuracy of each task. We also use a weighted average when calculating average forgetting:

$$\text{Average Forgetting} = \frac{1}{\sum_{k=1}^{K} \sum_{t=1}^{T-1} n_k^t} \sum_{k=1}^{K} \sum_{t=1}^{T-1} \max_{i \in \{1,...,T-1\}} (a_k^{i,t} - a_k^{T,t}) * n_k^t. \tag{22}$$

## C.4 Optimization

The Adam optimizer is employed for training all models. For all experiments except for CIFAR100, a learning rate of 1e-4 is utilized, with a global communication round of 60, and local iteration of 100. We set learning rate as 1e-3, the global communication round as 40, and local iteration as 400 for CIFAR100. Other parameters include $weight decay = 1e-5, beta1 = 0.9, beta2 = 0.999$. For training, a mini-batch size of 64 is adopted. The number of generated samples in an iteration aligns with this mini-batch size. We report the mean and standard deviation of each experiment, conducted five times with different random seeds.

## C.5 Model Architectures

In the case of CIFAR100, we utilize the feature extractor of a ResNet-18 [47] as $h_a$ and $h_b$ comprises two FC layers, both with 512 units. For other datasets we adopt a three-layer CNN followed by an FC layer with 512 units as $h_a$. The channel numbers of the convolutional layers are $[64, 128, 256]$. And $h_b$ is represented by an FC layer. The outputs of $h_a$ belong to $\mathbb{R}^{512}$. All the FC layers employed in the architectures consist of 512 units. The convolutional layers and FC layers are followed by a Leaky ReLU layer. Another FC layer serves as $h_c$ and operates as the classification head.

## C.6 Devices

In the experiments, we conduct all methods on a local Linux server that has two physical CPU chips (Intel(R) Xeon(R) CPU E5-2640 v4 @ 2.40GHz) and 32 logical kernels. All methods are implemented using Pytorch framework and all models are trained on GeForce RTX 2080 Ti GPUs.

# D  Additional Experimental Results

## D.1  More Complex Scenario

We conduct experiments on CIFAR100 in a more challenging setting. We randomly sample 10 classes among 100 classes of CIFAR100 as a task for each of the 8 clients, and there are 6 tasks for each client ($K = 8, T = 6, C = 10$). For each class, we randomly sample 400 instances into the client dataset. Therefore, each client possesses more tasks with fewer samples per task.

As shown in Table 5, our method achieves the highest average accuracy among the evaluated approaches. While the CL approach emphasizes retaining knowledge from previous tasks and the traditional FCL approach focuses on centralized aggregation to ensure that client knowledge is utilized totally, these methods can sometimes have a negative influence by indiscriminately aggregating information. In contrast, our proposed method utilizes decentralized federated aggregation to form client coalitions through dynamic cooperative learning. This approach aggregates clients with similar tasks, mitigating forgetting within local coalitions, especially when data heterogeneity among clients is significantly strong. Therefore, compared to established baselines, our method achieved the highest average task test accuracy.

## D.2  Communication Cost

To reduce communication overhead, we cache the model information from the previous aggregation round on both the client and the third party. This allows gradient information to be calculated by model differences, so only model parameters need to be transmitted in each communication round.

We list the communication cost in Table. 6 of different methods across four datasets. C2S represents client-to-server cost, S2C is server-to-client cost. The results demonstrate that DCFCL achieves optimal communication efficiency in all datasets, matching the performance of the most basic FedAvg and FedProx methods while significantly outperforming improved approaches that require additional communication overhead (such as SCAFFOLD and CFL, which typically double the communication volume in the C2S direction). It is particularly noteworthy that although methods like ACGAN and FedCIL enhance model performance by incorporating generative models, they all introduce varying degrees of increased communication costs. In contrast, DCFCL ensures performance improvements while completely avoiding additional communication burdens.

Table 6: The client to server(C2S) and sever to client(S2C) communication cost(GB) during the whole training process.

| Model | CIFAR100 | | EMNIST-LTP | | EMNIST-shuffle | | MNIST-SVHN-F | |
|---|---|---|---|---|---|---|---|---|
| | C2S | S2C | C2S | S2C | C2S | S2C | C2S | S2C |
| FedAvg | 10.400 | 10.400 | 4.056 | 4.056 | 4.056 | 4.056 | 7.260 | 7.260 |
| FedProx | 10.400 | 10.400 | 4.056 | 4.056 | 4.056 | 4.056 | 7.260 | 7.260 |
| SCAFFOLD | 20.800 | 10.400 | 8.112 | 4.056 | 8.112 | 4.056 | 14.520 | 7.260 |
| CFL | 20.800 | 10.400 | 8.112 | 4.056 | 8.112 | 4.056 | 14.520 | 7.260 |
| Per-FedAvg | 10.400 | 10.400 | 4.056 | 4.056 | 4.056 | 4.056 | 7.260 | 7.260 |
| PODNet+FedAvg | 20.800 | 10.400 | 8.112 | 4.056 | 8.112 | 4.056 | 14.520 | 7.260 |
| PODNet+FedProx | 20.800 | 10.400 | 8.112 | 4.056 | 8.112 | 4.056 | 14.520 | 7.260 |
| ACGAN+FedAvg | 10.523 | 10.400 | 4.093 | 4.056 | 4.093 | 4.056 | 7.440 | 7.260 |
| ACGAN+FedProx | 10.523 | 10.40 | 4.093 | 4.056 | 4.093 | 4.056 | 7.440 | 7.260 |
| FedCIL | 20.800 | 10.400 | 8.112 | 4.056 | 8.112 | 4.056 | 14.520 | 7.260 |
| AF-FCL | 20.800 | 10.400 | 8.112 | 4.056 | 8.112 | 4.056 | 14.520 | 7.260 |
| AFCL | 10.420 | 10.400 | 4.062 | 4.056 | 4.062 | 4.056 | 7.260 | 7.260 |
| FPPL | 10.420 | 10.400 | 4.062 | 4.056 | 4.062 | 4.056 | 7.260 | 7.260 |
| **DCFCL** | **10.400** | **10.400** | **4.056** | **4.056** | **4.056** | **4.056** | **7.260** | **7.260** |

## D.3  Mitigation of Catastrophic Forgetting

We compare the forgetting rate in Table. 7 to further demonstrate the effectiveness. The results clearly demonstrate DCFCL's superior performance in mitigating catastrophic forgetting, achieving

Table 7: The average forgetting rate $(\%)$ on 4 datasets.

| Model | CIFAR100 | EMNIST-LTP | EMNIST-shuffle | MNIST-SVHN-F |
|---|---|---|---|---|
| FedAvg | $8.6_{\pm 0.9}$ | $24.0_{\pm 0.6}$ | $9.6_{\pm 0.9}$ | $25.6_{\pm 0.6}$ |
| FedProx | $8.4_{\pm 0.6}$ | $23.8_{\pm 0.7}$ | $8.1_{\pm 0.6}$ | $24.9_{\pm 0.7}$ |
| SCAFFOLD | $8.2_{\pm 0.7}$ | $19.2_{\pm 0.3}$ | $8.2_{\pm 0.7}$ | $22.1_{\pm 0.3}$ |
| CFL | $8.9_{\pm 0.8}$ | $19.8_{\pm 0.6}$ | $9.4_{\pm 0.6}$ | $24.4_{\pm 0.8}$ |
| Per-FedAvg | $8.7_{\pm 0.7}$ | $19.4_{\pm 0.5}$ | $7.8_{\pm 0.6}$ | $21.9_{\pm 0.7}$ |
| PODNet+FedAvg | $8.6_{\pm 0.6}$ | $15.5_{\pm 0.7}$ | $7.3_{\pm 0.9}$ | $21.3_{\pm 0.3}$ |
| PODNet+FedProx | $7.5_{\pm 0.9}$ | $14.3_{\pm 1.2}$ | $6.0_{\pm 0.7}$ | $20.6_{\pm 0.8}$ |
| ACGAN+FedAvg | $6.4_{\pm 0.7}$ | $14.3_{\pm 0.5}$ | $6.5_{\pm 0.6}$ | $20.0_{\pm 0.8}$ |
| ACGAN+FedProx | $6.2_{\pm 0.4}$ | $12.4_{\pm 0.7}$ | $6.1_{\pm 0.5}$ | $19.7_{\pm 0.4}$ |
| FedCIL | $6.5_{\pm 0.2}$ | $10.4_{\pm 0.4}$ | $6.4_{\pm 0.8}$ | $19.7_{\pm 0.8}$ |
| AF-FCL | $4.9_{\pm 0.9}$ | $\mathbf{7.9}_{\pm 0.4}$ | $\mathbf{4.2}_{\pm 1.4}$ | $7.5_{\pm 0.8}$ |
| AFCL | $6.3_{\pm 0.5}$ | $10.5_{\pm 0.9}$ | $5.7_{\pm 1.1}$ | $11.3_{\pm 0.5}$ |
| FPPL | $6.9_{\pm 0.6}$ | $11.6_{\pm 0.3}$ | $7.4_{\pm 0.9}$ | $13.2_{\pm 0.2}$ |
| DCFCL | $\mathbf{4.7}_{\pm 0.9}$ | $8.9_{\pm 0.6}$ | $\mathbf{4.2}_{\pm 0.8}$ | $\mathbf{6.9}_{\pm 0.7}$ |

the lowest forgetting rates on 3 datasets. This represents reduction compared to baseline methods like FedAvg and FedProx. DCFCL's dynamic collaboration mechanism achieves significantly better retention without requiring additional memory buffers or complex architectural modifications. These consistent improvements across diverse datasets underscore DCFCL's robustness in preserving learned knowledge while accommodating new information.

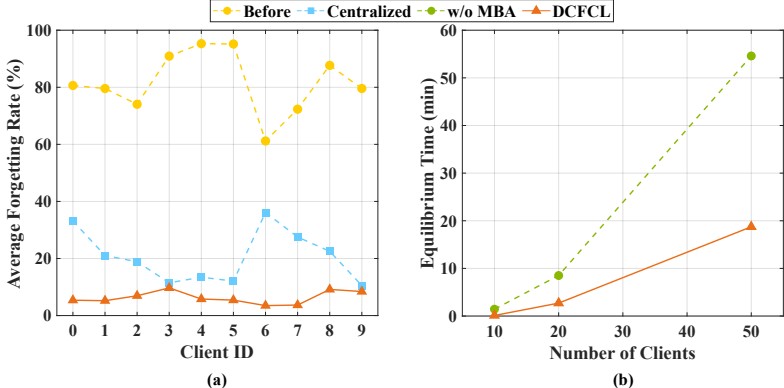

Figure 7: Comparison of average forgetting of each client on MNIST-SVHN-F(left). The equilibrium achieving time when the number of clients increases(right).

We also list forgetting mitigation of each client to illustrate the benefits of cooperation comparing no aggregation (local), centralized aggregation and decentralized (DCFCL) methods, as shown in Fig. 7(a). The local method (yellow) show a high forgetting rate of 60%-95%. After adopting centralized aggregation (blue), the forgetting rate significantly decrease to 10%-36%, indicating that the aggregation between clients can promote the knowledge recall of different clients respectively, but there is still room for optimization. The decentralized dynamic cooperation method (orange) demonstrate better results, stably controlling the forgetting rate below 10% (3.49%- 9.70%). It is particularly worth noting that DCFCL maintains the lowest and most stable forgetting rate on all clients, significantly reducing the differences in forget rates among clients.

## D.4 Computation Cost

We present Fig. 7(b) to show the computation time of equilibrium as the client number increases, comparing it to the method without merge-blocking algorithm (MBA). As the number of clients grows from 10 to 50, the conventional approach without MBA shows exponential time escalation, highlighting scalability issues. In contrast, DCFCL maintains polynomial time complexity, with computation times increasing only from 0.116 min to 18.733 min, with the gap widening as the system scale grows. This demonstrates DCFCL's advantage in large-scale deployments.

Table 8: The run-time consumption comparisons $T(min)$ on 4 datasets.

| Model | CIFAR100 | EMNIST-LTP | EMNIST-shuffle | MNIST-SVHN-F |
|---|---|---|---|---|
| FedAvg | 238 | 22 | 21 | 29 |
| FedProx | 246 | 26 | 24 | 32 |
| SCAFFOLD | 265 | 38 | 37 | 45 |
| CFL | 294 | 34 | 35 | 47 |
| Per-FedAvg | 287 | 32 | 28 | 32 |
| PODNet+FedAvg | 252 | 35 | 34 | 49 |
| PODNet+FedProx | 253 | 37 | 39 | 51 |
| ACGAN+FedAvg | 312 | 85 | 81 | 149 |
| ACGAN+FedProx | 315 | 89 | 82 | 152 |
| FedCIL | 322 | 93 | 90 | 172 |
| AF-FCL | 302 | 62 | 60 | 126 |
| AFCL | 277 | 34 | 32 | 44 |
| FPPL | 253 | 32 | 31 | 45 |
| DCFCL | 294 | 39 | 39 | 48 |

Table. 8 compares runtime performance across four datasets. DCFCL shows competitive runtime (48-294 minutes), similar to CFL and SCAFFOLD. Generative methods (ACGAN and FedCIL) incur higher overhead (up to 322 minutes on CIFAR100), while traditional methods like FedAvg and FedProx are faster (32-238 minutes) but may sacrifice performance. DCFCL strikes a balance between efficiency and learning ability, with moderate runtime costs across datasets.

