# OpenReview forum: "Decentralized Dynamic Cooperation of Personalized Models for Federated Continual Learning"
_NeurIPS.cc/2025/Conference — NeurIPS 2025 poster_

### Official Review · Reviewer_1yUZ · 2025-06-03

**Clarity:** 2
**Significance:** 2
**Originality:** 2
**Rating:** 4
**Confidence:** 4

**Summary:**

This paper proposes a decentralized dynamic cooperation framework where clients form adaptive coalitions for personalized learning, leveraging affinity games and new algorithms to achieve superior performance over existing baselines. The proposed framework is partially validated on small-scale experiments, without providing relevant theoretical guarantees.

**Questions:**

(a) Is the concept of decentralization introduced in this paper merely an arbitrary grouping of clients? This does not seem to capture the true essence of decentralization. Although some explorations of push-sum-like topologies have suggested that dynamic links might offer better alternatives, the analysis of such schemes is still fundamentally based on the investigation of consensus constraints or spectral gap properties. It appears that the present paper has misused the concept of decentralization. The authors are requested to clarify the similarities and differences between their approach and traditional decentralization. In my opinion, these concepts should not be conflated, and the paper seems to require substantial revision.

(b) What is the overall optimization objective of this paper? At present, it seems that only the local optimization form on the client side has been defined, which appears to be completely unrelated to personalized federated learning. I would like to know the specific form of the global optimization objective.

(c) The content in Section 3.3 does not appear to be original. The authors should cite relevant literature and clearly indicate which parts of this section are their own contributions.

(d) Please provide the computational complexity or training wall-time curves for both the baseline methods and the proposed method. The approach presented in this paper involves optimizing with four loss functions, which results in significantly higher computational complexity compared to the original methods. However, the paper does not report any metrics related to training cost. Based on the current description, the proposed method may be extremely expensive in terms of computational resources.

**Ethical Concerns:**

["NO or VERY MINOR ethics concerns only"]

**Final Justification:**

The optimization problem in this paper is not well described. After multiple discussions with the authors, I have come to understand the actual optimization objective and the corresponding solution approach. The game-theoretic method for FL aggregation proposed in this work is not novel, but the current solution approach is correct. Its advantage lies in its ability to strictly guarantee balance on each communication round, while its drawback is that the computational complexity grows exponentially with the number of clients. At present, I have no further concerns and will raise my score to 4.

**Limitations:**

yes

**Quality:**

2

**Strengths And Weaknesses:**

Strengths:

The paper provides a detailed introduction to the proposed dynamic collaboration process, offering a reference model for the federated continual learning framework. It also incorporates grouping schemes based on similarities in gradients and model parameters, providing practical guidance for engineering applications.

Weaknesses:

The classification of the basic settings of federated learning in this paper is incorrect. For example, in the paragraph around line 39, when discussing performance comparisons under different settings, the paper conflates general FL, PFL, and even DFL. The goal of general FL is to collaboratively train a single global model across multiple nodes, while the objective of PFL is to train multiple local models. DFL refers to a communication method induced by different topology; it can be used to accomplish both FL and PFL training. I suggest the authors revise the entire manuscript, or at least clarify the classification and task objectives related to FL throughout the paper.

The motivation of this paper is not clearly stated. Simply adopting aggregation to avoid catastrophic forgetting, or using decentralized aggregation to improve the performance of models within groups, makes the work appear incremental. It is currently unclear what core challenge the submission aims to address. Could the authors further clarify, perhaps through mathematical formulations or other quantifiable characteristics, what specific problems this series of techniques is designed to solve?

The series of techniques proposed in this paper do not appear to be fully original, and they also lack theoretical analysis and guarantees.

The experimental setup is small and differs significantly from typical federated learning settings. Standard FL usually involves collaborative training with hundreds of clients, whereas the setup in this paper appears to be limited to only 8 to 10 clients. Thus it becomes much easier to achieve significant control through data partitioning or task allocation. However, in practice, when hundreds of clients are involved in training, certain similarity measures and training methods may not be as effective as they are in small-scale experiments.

---

> ### Author Rebuttal · Authors · 2025-07-31
>
> ## Reviewer 1yUZ
> We would like to thank the reviewer for the insightful comments. Below are our responses to the comments in **Weaknesses and Questions**.
>
> ---
>
> **Respond to W1**
>
> **Answer:** We thank the reviewer for their suggestions. Personalized FCL enables clients to maintain individual task histories and learn customized models through cooperation [1,2]. Unlike centralized approaches, decentralized learning eliminates the need for a central server [2,3], better aligning with our goal of optimizing individual client performance (Eq.1) rather than global objectives. As discussed in Section 1 (Paragraphs 2-3), decentralized topologies are more suitable for personalized FCL. The comparison of CFL, Per-FedAvg, and centralized methods (Section 1, line 39) demonstrates that selective cooperative aggregation is essential. We apologize for any lack of clarity and will refine these definitions in the revised manuscript.
>
> ---
> **Respond to W2**
>
> **Answer:** We thank the reviewer for arguing this point. Our work addresses the key challenge of ***how to achieve personalized continual learning in federated systems with strong spatio-temporal catastrophic forgetting*** as stated in Section 1.
>
>
>
> 1. FCL Challenges
> FCL introduces two key challenges:
> (1) Local temporal catastrophic forgetting from sequential task learning
> (2) Spatial catastrophic forgetting during aggregation of heterogeneous models
>
> 2. Assumptions and Initial Evidence
> We first establish that federated aggregation can mitigate forgetting when models share homogeneous knowledge (Fig.1a-b), explaining central server architectures [4,5].
>
> 3. Limitations of Complete Aggregation
> Complete aggregation proves suboptimal under spatial heterogeneity due to knowledge interference (Fig.1c), showing aggregation efficacy depends on methodology.
>
> 4. Connection to Personalized FL
> While methods like ClusterFL (decentralized) and Per-FedAvg handle spatial heterogeneity, they fail to address continual learning challenges (lines 45-49).
>
> This motivates our DCFCL framework featuring dynamic grouping to:Handle continuous task arrivals，Prevent spatio-temporal forgetting，Maintain decentralized personalization
>
>
>
> ---
>
> **Respond to W3**
>
> **Answer:**We thank the reviewer for raising this point. We are the first to propose a dynamic grouping strategy for personalized model learning in FCL which is entirely original. We would like to clarify our contributions as follows:
> 1.**Theoretical Foundation**
> Section 3.1 formally establishes the problem of personalized FCL in decentralized topologies. Section 3.2 introduces our novel dynamic grouped aggregation approach for model updates. Section 3.3 applies cooperative game theory to determine optimal groupings, with the cooperative equilibrium as our target solution.
>
> 2.**Core Innovations in Chapter 4**
> We extend static cooperative games to dynamic settings to achieve dynamic grouping. While inspired by coalitional affinity games for relationship modeling, all algorithmic components are original:(1) Knowledge distillation (creatively applied) maintains feature consistency for cooperator identification
> (2) Our benefit table framework uses:
> Original overall similarity metric,Coalitional affinity principles
> (3) We created:Merge-blocking algorithm,Dynamic cooperative evolution algorithm,to obtain Dynamic Cooperative Equilibrium
>
> 3.**Theoretical Analysis**
> Appendix A contains our original arguments about coalitional affinity game applications. Extensive experiments validate:KD,OS,CE.All aspects from problem formulation to equilibrium analysis represent original FCL contributions, except: affinity-based relationship concept,KD's core mechanism.
> We believes that this can serve as theoretical analysis and guarantee to a certain extent. If we have any misunderstandings, please do not hesitate to question us. We will do our best to clear up your doubts.
>
> ---
>
> **Respond to W4**
>
> **Answer:** Our experimental results (Fig.7, Appendix D.4) demonstrate that while conventional equilibrium computation exhibits exponential time growth as client numbers increase from 10 to 50, DCFCL maintains near-linear scalability - a performance gap that widens significantly with system scale. The proposed Merge-Blocking Algorithm achieves substantial efficiency gains, including faster equilibrium formation and fewer iterations, while significantly reducing both time complexity  and communication costs  compared to baseline methods.
>
> |Number of clients|10|20|30|40|50|
> |-|-|-|-|-|-|
> |Time (min)|0.116|2.715|8.279|12.526|2.715|
> |Iterations|2|2|3|3|5|
> |Time (min)-w/o MBA|1.438|8.504|24.733|39.293|54.596|
>
> We acknowledge the combinatorial complexity challenge, though our current focus remains on moderate-scale federations to study fundamental collaboration dynamics. While computational cost isn't our primary focus given the priority of precise collaborator identification, our framework demonstrates strong performance/speed trade-offs versus alternatives ([6]'s loss distance, [7]'s FC-based, [8]'s gradient-based). Like most group-based FL methods, pairwise similarity computations inherently limit scalability for very large systems. To address scalability we plan to use:1.Sinkhorn Approximation: For large client counts, we employ Sinkhorn's entropy-regularized optimal transport [9] to efficiently approximate similarity matrices through iterative refinement. 2.Localized Assessment: Clients compute similarities only with Top-K neighbors [10] or historical collaborators.3.Sparse Matrix Construction: We retain only high-similarity edges in the similarity graph.4.Algorithmic Optimization: Our merge-blocking algorithm processes large coalitions first and only traverses non-zero matrix entries.
>
> In summery, we will leave these promising extensions to future work and clarify this in revision.
>
> ---
>
> **Respond to Q1**
>
> **Answer:** We clarify our decentralized FCL approach through four key aspects:
>
> Decentralization Advantage: Unlike centralized aggregation, our decentralized topology better enables personalized model formation, achieving client-satisfactory group aggregation without central server involvement.
>
> Cooperative Game Foundation: Our dynamic group strategy is theoretically grounded in cooperative games, where rational clients form performance-optimizing coalitions while excluding non-contributors - fundamentally differing from clustering methods.
>
> Behavioral Personalization: Client autonomy in alliance formation/betrayal embodies personalization, with the decentralized structure naturally aligning with cooperative game theory's agent-centric dynamics.
>
> Traditional Comparison: While traditional decentralized methods ([11]'s gossip-SGD, [12]'s doubly-stochastic weights) enforce global consensus ([13]), they falter in non-IID settings ([14]). Like [15,16], we adopt decentralization for personalized modeling, but uniquely integrate it with continual learning.
>
> ---
>
> **Respond to Q2**
>
> **Answer:** Thank you for your questioning. Our overall optimization goal is specific to each client, which is precisely what personalized FL defines[17,18], as described in Section 3.1 Problem Setup. At each task phase $t$, model parameter of client $k$ is $θ_k^t$ , and optimization goal is foreach client. Because it is personalized model and adopts a decentralized structure, we do not have a global optimization objective. This is the main difference from the centralized method, as the latter may optimize a global objective.
>
> ---
>
> **Respond to Q3**
>
> **Answer:** Thank you for your questioning. In the revised version, we will cite the article of cooperative game theory.
>
> ---
>
> **Respond to Q4**
>
> **Answer:** We provide comprehensive runtime comparisons with all baselines across datasets in Appendix D.4 (full training wall-time), along with time complexity analysis versus non-optimized cooperative strategies in Appendix B.4. Our method avoids four-loss optimization, and we detail communication costs (Appendix D.2) and computation costs (Appendix D.4). We apologize for any unclear presentation and will clarify these analyses in revision.
>
>
>
> [1] J. Xie, FedMeS: Personalized Federated Continual Learning Leveraging Local Memory.
>
> [2] S. Li, Learning to Collaborate in Decentralized Learning of Personalized Models,.
>
> [3] S. Kharrat, DPFL: Decentralized Personalized Federated Learning.
>
> [4] Yoon, J. Federated continual learning with weighted inter-client transfer.
>
> [5] Shenaj, D.. Asynchronous federated continual learning.
>
> [6] M. Zhang, "Personalized Federated Learning with First Order Model Optimization.
>
> [7] Wu, S.. PFedCS: A Personalized Federated Learning Method for Enhancing Collaboration among Similar Classifiers.
>
> [8] Sattler, F.. Clustered Federated Learning: Model-Agnostic Distributed Multitask Optimization Under Privacy Constraints..
>
> [9] Sinkhorn, R. A relationship between arbitrary positive matrices and doubly stochastic matrices.
>
> [10] S. Li, T. Zhou, X. Tian and D. Tao, "Learning to Collaborate in Decentralized Learning of Personalized Models.
>
> [11] Michael Blot, . Gossip training for deep learning.
>
> [12] Zhanhong . Collaborative deep learning in fixed topology networks.
>
> [13] Xiangru Lian, . Can decentralized algorithms outperform centralized algorithms? a case study for decentralized parallel stochastic gradient descent.
>
> [14] Kevin Hsieh, The non-iid data quagmire of decentralized machine learning.
>
> [15] S. Li, "Learning to Collaborate in Decentralized Learning of Personalized Models.
>
> [16] P. Vanhaesebrouck, , "Decentralized Collaborative Learning of Personalized Models over Networks," arXiv preprint arXiv:1610.05202, 2017.
>
> [17] S. Li, "Learning to Collaborate in Decentralized Learning of Personalized Models," 2022 IEEE/CVF Conference on Computer Vision and Pattern Recognition.
>
> [18] Wu, S., . PFedCS: A Personalized Federated Learning Method for Enhancing Collaboration among Similar Classifiers. .

---

> > ### Comment · Reviewer_1yUZ · 2025-08-02
> > **Response to authors**
> >
> > I sincerely appreciate the authors’ detailed responses to my questions. However, it seems that the authors may have misunderstood part of my concerns, so I still have some doubts regarding this issue.
> >
> > **objective**
> >
> > The authors state that the goal of PFL is simply to optimize local models, but this is actually not accurate. If its objective were merely to $\min_{\theta_k} L_k(D_k, \theta_k)$, then there would be no need for any communication at all, and joint learning across multiple clients would not be necessary, since the optimal solution would simply be the local model. Here, I would like to emphasize that the typical objective of PFL is to $\min_{[\theta_0, \theta_1, \cdots, \theta_k]}\frac{1}{k}\sum_{i}L_i + \text{reg}(\cdot)$. The necessity of joint training in PFL arises because specially designed regularization terms make the local optimum for each individual model no longer the optimal solution to the overall problem. Therefore, clients must collaborate through joint training to achieve the true optimum.
> >
> > The reason I asked this question is that I did not understand at all the maximization of benefits and minimization function defined in line 135, which is
> > $$
> > u_{k}(s_\star^\tau) = \max_m u_k(s_m^\tau) = \min_{\theta_k^\tau} L_k^\tau(D_k^\tau, \theta_k^\tau), \theta_k^\tau=\cdots.
> > $$
> > First of all, how are this maximization problem and minimization problem equivalent? Their optimization objectives are not the same. Why can finding the index $m$ that maximizes the objective be solved by minimizing the local model? The solution to your minimization problem is a $d$-dimensional vector (parameters), but the solution to your maximization problem is the index $m$—how can these two possibly be equivalent? Secondly, your minimization problem is defined as solving for the local model $\theta_k$, but later $\theta_k$ appears to have an additional definition. I find this optimization objective extremely confusing, to the point that I cannot currently understand what the background of this problem setup is actually trying to optimize. I hope the authors can provide a clear definition of the optimization objective, so that I can know whether the optimization method used later is correct, or whether the proposed method truly solves the optimization objective itself. Strictly speaking, the authors need to provide a rigorous formulation of the PFL objective, including a clear definition of the regularization term. However, I hope that at the very least, the authors can clearly define the objective itself in their next response. At present, I find the definition of this problem highly confusing, and I am certain that at least the section concerning the optimization objective requires major revision.
> >
> > **computational complexity or training wall-time**
> >
> > I noticed the time-related experiments in the authors’ appendix during my first review, but that was not the focus of my question. Since the proposed method involves combinatorial optimization and game-theoretic aspects, in my understandings, the aggregation step essentially requires solving an additional optimization problem. What I would like to know is: What is the computational complexity and time cost of solving this problem, and how do they scale as the number of clients increases? The complexity of combinatorial problems is typically related to the size of the combinations, and in the worst case, can even be exponential (factorial) in the number of clients. In practice, PFL training usually involves hundreds or even thousands of clients. Therefore, I am interested in the per-iteration or per-communication-round complexity of your method, and how its running time compares to other methods as the number of participating clients increases. In other words, can the proposed method scale to hundreds or thousands of clients without incurring excessive time costs?
> >
> > Other issues may be considered minor; the two points above are my main concerns. If there is any misunderstanding on my part, I hope the authors can provide a detailed explanation.

---

> > > ### Author Response · Authors · 2025-08-04
> > >
> > > We sincerely thank for reviewer's response. Here are our answer to 2 **questions**.
> > >
> > > **R1**
> > >
> > > ***Answer:*** Thank you. We broke down it into 2 parts.
> > >
> > > 1. **Objective Rationality**:
> > > Our optimization objective is $\arg\min_{\theta_k^{\tau}} E_{k} \left[ L_{k}(\theta_{k}^{\tau}) \right]=\arg\min_{[\theta_i^{\tau-1}|i\in S]}E_{k} \left[ L_{k}\left(\sum_{i \in S} \alpha_i \left( \theta_i^{\tau-1} + \Delta \theta_i^{\tau-1} \right)\right) \right]$, it is decided under model update: $\theta_{k}^{\tau}=\sum_{i \in S} \alpha_i \left( \theta_i^{\tau-1} + \Delta \theta_i^{\tau-1} \right)$. It cannot be replaced by your expression for 2 reasons:
> > > (1) As categorized in [1], PFL employs 2 strategies: (i) Global Model Personalization - training a shared global model followed by local adaptation, (ii) Learning Personalized Models - directly building customized models through modified aggregation. We adopts **(ii)**, optimizing each client's local loss, whereas expression you provided should be classified to **(i)**.
> > > (2) If we haven't misunderstood,  "reg(.)" can be generalized by $l_{reg}\left(\theta_k;w_k\right)$, where $w_k$ is aggregated model for coalition $S$ which $k$ belongs to. This is regularized local loss approach in Strategy (i) [1]. Difference from ours is: instead of adding regularization terms to make $\theta_k$ approach $w_k$, we directly update $\theta_k$ with $w_k$, making $w_k$ become an integral part of new model.
> > >
> > > Our objective simultaneously addresses: local model optimization ($\theta_k^{\tau-1}$), and coalition-level optimization ($S$). This dual focus distinguishes our approach from conventional local model optimization and highlights joint learning.
> > >
> > > 2. **Benefit Rationality**:
> > > (1)**eq 1**: indicates we aim to find $s_m^\tau$, its partition $\pi$ maximizes client $k$'s benefit.
> > > (2)**eq 2**: indicates criterion for maximizing benefits is error (loss) of aggregated model is minimum on $k$'s testset after coalition [2]. Therefore, $u_k(s_m^\tau):=-L_k(\theta_k^\tau)$，$\theta_k^\tau$ is aggregated model in $S$，$S\in\pi(s_{m}^{\tau})$. By adjusting index $s_m^\tau$ to find $\theta_k^\tau$, process is the same. Furthermore, $\max u_k(s_m^\tau)=\max(-L_k(\theta_k^\tau))=\min(L_k(\theta_k^\tau))$. Thanks a lot for your point! In the revised version, we will change $L_k(\theta_k^\tau)$ to $L_k(\theta_k^\tau, D_k^{val})$ to prevent misunderstandings.
> > >
> > > **R2**
> > >
> > > ***Answer:*** Thank you. Since there are few PFL algorithms using group aggregation, for fairness, we select representative ones for evaluation.
> > >
> > > | Algorithm | Benefit Calculation | Coalition Formation | Coalition's Number | Rational Optimal Solution | Dynamic Coalition |
> > > |:-:|:-:|:-:|:-:|:-:|:-:|
> > > | ClusterFL [3] | $O(K^2)$ | $O(K^4)$ | 2 | × | × |
> > > | FedGroup [4] | $O(KM)$ | $O(KM^2 + TK^2M)$ | $M$ | × | × |
> > > | Coalitional FL [5] | $O(2^{K-1})$ | $O(\max (K^3, K2^{l_{\max}}))$ | unlimited | × | × |
> > > | pFedSV [6] | - | $O(k! KN)$ | $K$ | × | × |
> > > | DCFCL | $O(K^2)$ | $O(K2^K)$ | unlimited | √ | √ |
> > >
> > > **ClusterFL** quantifies benefits through pairwise similarity and employs Optimal Bipartition Algorithm to minimize inter-group similarity; **FedGroup** decomposes all weights via SVD into $M$ vectors and applies K-means++ clustering over $T$ iterations; **Coalition FL** utilizes EMD-based linear combinations of data distributions with Accelerated Device Coalition Formation Algorithm (whose complexity matches ours when $l_{max}=K$, the maximum number of clients), **pFedSV** forms coalitions for each client via top-$k$ Shapley values at $O(k!KN)$ complexity for $N$ test samples. Compared with them, our algorithm: (1) provides game-theoretically optimal solution for rational clients, though with increased complexity compared to clustering ones [3,4]; (2) dynamic, scale-unlimited coalition better adapts to continual learning, as evidenced by superior performance; (3) while maintaining computational efficiency through coalition affinity game and structured assumptions (additive/symmetric benefits), we quantify benefits for each client - a feature shared only with **Coalition FL** - providing a reliable idea for incentives and benefit allocation.
> > >
> > > While our method achieves optimal performance at relatively small scale, practical deployment for large scale necessitates approximating solution, which sacrifices theoretical optimality for computational feasibility, thereby becoming a performance-cost tradeoff.
> > >
> > >
> > > [1] A. Z. Tan, Towards Personalized Federated Learning.
> > >
> > > [2] K. Donahue, Optimality and Stability in Federated Learning: A Game-theoretic Approach.
> > >
> > > [3] F. Sattler, Clustered Federated Learning: Model-Agnostic Distributed Multitask Optimization Under Privacy Constraints.
> > >
> > > [4] M. Duan,FedGroup: Efficient Federated Learning via Decomposed Similarity-Based Clustering.
> > >
> > > [5] N. Zhang, Coalitional FL: Coalition Formation and Selection in Federated Learning With Heterogeneous Data.
> > >
> > > [6] L. Wu, A Coalition Formation Game Approach for Personalized Federated Learning.

---

> > > > ### Comment · Reviewer_1yUZ · 2025-08-04
> > > >
> > > > Thanks for the authors’ explanation. However, I believe that there are indeed significant issues with the notations. Too many variables are introduced into the optimization objective; for example, including the aggregated model as a local update variable in the optimization objective creates considerable confusion for readers. This problem is particularly prominent in Section 3.3, in the definition of the method and objective.
> > > >
> > > > I suggest that the authors reconstruct the game process using a functional representation, rather than substituting it in the form of parameter updates, as this makes it impossible to clearly specify the optimization objective. In the current description, the indices for “min” and “max” are completely different variables, which cannot be equivalent in mathematical terms.
> > > >
> > > > The complexity table in the response from R2 matches my previous understanding, and this should be the main limitation of the proposed algorithm. I hope the authors can include this table in the main text to clearly illustrate both the strengths and weaknesses of the algorithm. Additionally, I have a further question. I noticed that the authors mentioned the possibility of using approximation methods to reduce complexity. Are there any specific designs or strategies for such approximate solutions?

---

> > > > > ### Author Response · Authors · 2025-08-06
> > > > >
> > > > > We sincerely thank the reviewer for their detailed responses. Here are our responses to problems.
> > > > >
> > > > > **Respond to Variables**
> > > > >
> > > > > ***Answer:*** Thank you. In developing our DCFCL, we integrated multiple knowledge domains including FL, CL, cooperative games, knowledge distillation, and affinity relationships - each requiring mathematical modeling. While we followed established variable notations from prior works [1-5], the integration inevitably led to significant variables. We acknowledge your concern regarding potential confusion caused by using the aggregated model as an independent variable in optimization objective, so we plan to  reformulate Eq. (1) with more explicit notation such as
> > > > >
> > > > > \begin{equation}
> > > > > \arg\min_{\{\theta_k^{\tau-1},S|k \in S\}} E_{k} \left[ L_{k}(\sum_{i \in S} \alpha_i \left( \theta_i^{\tau-1} + \Delta \theta_i^{\tau-1} \right)
> > > > > ) \right]
> > > > > \end{equation}
> > > > >
> > > > > This shows that for a given client, others' models are fixed, reducing optimization variables to just two parts: local model and coalition. This avoids using parameter updates as objectives. We will also include a notation table in Appendix for reference.
> > > > >
> > > > > **Respond to Min-Max Fuction**
> > > > >
> > > > > ***Answer:*** Thanks a lot. Our orginal expression can be conducted by:
> > > > >
> > > > > (1) $\max_m u_k(s_m^\tau)=\max_{S|S \in \pi(s_m^\tau)}-L_k(\sum_{i \in S} \alpha_i  \theta_i^{\tau}, D_k^{val}))$
> > > > >
> > > > > Here, $u_k = -L_k$ holds, showing aggregated model's test loss on $k$ equals its benefit, with state $s_m^\tau$ mapped by $S$. Thus, equivalence is valid.
> > > > >
> > > > > (2)
> > > > > \begin{equation}
> > > > > \max_{S|S \in \pi(s_m^\tau)}-L_k(\sum_{i \in S} \alpha_i  \theta_i^{\tau}, D_k^{val})=\max_{\theta_k^\tau}-L_k(\theta_k^\tau,D_k^{val})=\min_{\theta_k^\tau}L_k(\theta_k^\tau,D_k^{val})
> > > > > \end{equation}
> > > > >
> > > > > Since $\sum_{i \in S} \alpha_i \theta_i^\tau = \theta_k^\tau$ and all client models remain fixed during aggregation, $\theta_k^\tau$ becomes a function of both fixed models and variable $S$. The one-to-one mapping between $\theta_k^\tau$ and $S$ implies the minimum loss can only be achieved by optimizing $S$, validating this equality.
> > > > >
> > > > > While this function is mathematically valid, we acknowledge it may confuse readers. We sincerely appreciate your insightful suggestion. In revision, we will present this complete process to avoid single-step equal substitutions.
> > > > >
> > > > > **Respond to Approximation**
> > > > >
> > > > > ***Answer:*** Thank you for your question. We will specifically design approximation method from 2 parts:
> > > > >
> > > > > 1. **For Benefit Calculation**:
> > > > > (1) **SVD**：decomposes models into small-scale directions by using Singular Value Decomposition (SVD), similarities for pair-client can be calculated by difference value of models and these directions to avoid directly calculation. ($O(K^2)->O(KM),M<<K$) .
> > > > > (2) **LSH**: Client locally calculates Locality-Sensitive Hashing encoding ($d=L$) of the models. Build hash table and precisely calculates models within same hash bucket (sharing at least 1 encoding) [6]. ($O(K^2)->O(KL),L<<K$)
> > > > > (3) **Nystrom**: samples a small number of client to form $m$ "anchor points", performs low-rank approximation on them, and calculates kernel matrix of anchor points and all models [7] to avoid directly calculation. ($O(K^2)->O(Km+m^3),m<<K$)
> > > > >
> > > > > 2. **For Coalition Formation**:
> > > > > (1) **Hierarchical Tree**: reformulate search structure using state transitions. Construct a coalition graph where edges represent coalition split operations, then convert to a hierarchical tree via DFS. Through recursive, search paths that depend on the graph structure can be limited, and the original state can only be re-divided into two new coalitions, reducing the search space [8]. e.g. for [{1,2,3}], next search can be [{1,2},{3}], [{1,3},{2}] and [{2,3},{1}], only to two slipt. Three-client aggregation builds upon two-client performance. By solving leaf nodes first and propagating upwards to parent nodes, we obtain suboptimal solutions in subspace.
> > > > > (2) **Graph Clustering**: Build client similarity graphs (nodes=clients, edges=similarity of pair-client), then apply threshold clustering to obtain initial partition, only perform merge-blocking on it to find equilibrium.
> > > > > (3）**Top-K SC**：based on merge-blocking，track top-k frequent coalitions as stable coalitions after an iteration, prune their members, then iteratively identify new stable coalitions in remaining space until equilibrium.
> > > > >
> > > > > [1] Yoon. Federated continual learning with weighted inter-client transfer.
> > > > >
> > > > > [2] McMahan. Communication-Efficient Learning of Deep Networks from Decentralized Data.
> > > > >
> > > > > [3] Konishi. Coalition formation as a dynamic process.
> > > > >
> > > > > [4] Usmanova. A distillation-based approach integrating continual learning and federated learning for pervasive services.
> > > > >
> > > > > [5] Simina. Coalitional affinity games and the stability gap.
> > > > >
> > > > > [6] Yang. Efficient Clustered Federated Learning by Locality Sensitive Hashing.
> > > > >
> > > > > [7] Williams. Using the Nyström Method to Speed Up Kernel Machines.
> > > > >
> > > > > [8] T. Rahwan. An improved dynamic programming algorithm for coalition structure generation.

---

> > > > > > ### Comment · Reviewer_1yUZ · 2025-08-06
> > > > > >
> > > > > > Thank you for authors' response. I do think it is necessary to carefully revise the notation system in this paper. Even when multiple research domains are involved, a clear problem definition helps readers understand the essence of the problem and the methodology used. After your explanations and reading your paper four times, I believe I have understood the core objective of this work. I will change my rating to positive. Thank you for your thorough and patient responses.

---

> > > > > > > ### Author Response · Authors · 2025-08-06
> > > > > > >
> > > > > > > **Many thanks for raising the score**
> > > > > > >
> > > > > > > Dear Reviewer,
> > > > > > >
> > > > > > > Thank you for your constructive comments and for taking the time to review our manuscript. We sincerely appreciate your feedback and are glad that our revisions addressed your concerns. We will add Complexity Comparison Table above and modify equations in revision. We are grateful for your precious support and insightful suggestions!
> > > > > > >
> > > > > > > Best Regards,
> > > > > > >
> > > > > > > Authors

---

### Official Review · Reviewer_ckij · 2025-06-29

**Clarity:** 3
**Significance:** 4
**Originality:** 3
**Rating:** 4
**Confidence:** 4

**Summary:**

This paper presents DCFCL, a decentralized framework for Federated Continual Learning where clients form dynamic coalitions via game theory to mitigate catastrophic forgetting. To efficiently estimate cooperation benefits without expensive trials, it uses a proxy metric called "overall similarity," which combines gradient and model similarity as $\cos(g_i, g_j) + \epsilon \cdot \cos(\theta_i, \theta_j)$. A merge-blocking algorithm is then used to find a stable coalition equilibrium.

**Questions:**

1.The $O(K2^K)$ complexity makes your method impractical for large-scale FL. How do you propose to solve this for K > 100? Are there viable approximation schemes?
2.Your "overall similarity" proxy is a heuristic. Can you provide a concrete example (synthetic or otherwise) where it fails and leads to a sub-optimal coalition compared to the ground-truth benefit?
3.Your framework describes a complex dynamic system. Can you provide a proof or at least a strong argument for the convergence of the overall client model training process across tasks? Without it, the stability of the entire learning process is in question.
4.Justify the decision to re-evaluate coalitions every round. How does it compare in terms of final accuracy versus total computation time against a less frequent update schedule (e.g., once per task)?

**Ethical Concerns:**

["NO or VERY MINOR ethics concerns only"]

**Limitations:**

1.The exponential computational complexity ($O(K2^K)$) is a hard barrier to scalability in systems with many clients.
2.The framework introduces several critical hyperparameters (e.g., $\epsilon$ for similarity weighting, $\lambda$ for knowledge distillation) whose optimal settings can be highly sensitive to the dataset and task sequence, and the paper offers no adaptive mechanism for tuning them.
3.The reliance on a trusted third party for model collection and computation is still a form of centralization with potential privacy risks.

**Quality:**

3

**Strengths And Weaknesses:**

Strengths
The work tackles a significant and difficult problem in FCL concerning dynamic heterogeneity. The core idea of using game-theoretic dynamic coalitions is novel and represents a solid contribution. Furthermore, the "overall similarity" proxy is a clever method to bypass the prohibitive computational cost of evaluating all possible coalitions directly. The experimental validation is also a major strength, featuring a comprehensive set of baselines, well-chosen datasets, and convincing ablation and communication cost analyses.

Weaknesses
1.The primary weakness is the method's scalability. The core algorithm's $O(K2^K)$ complexity is a fatal flaw for real-world federated learning with a large number of clients (K), a point the paper largely glosses over.
2.The benefit function is a pure heuristic with no analysis of its failure modes; it's unclear when the $\cos(g) + \epsilon \cdot \cos(\theta)$ proxy might be misleading.
3.The paper lacks an overall convergence analysis. While the merge-blocking algorithm finds a static equilibrium for a given round, there's no proof that the entire FCL process, with dynamically evolving coalitions, converges to a meaningful solution.
4.The decision to reform coalitions every single round seems inefficient and lacks justification compared to less frequent, event-triggered updates.

---

> ### Author Rebuttal · Authors · 2025-07-31
>
> ## Reviewer ckij
> We would like to thank the reviewer for the positive and insightful comments. Below are our responses to the comments in **Weaknesses and Questions**.
>
> ---
>
> #### ***Weakness 1& Question 1
>
> **Answer:** We thank the reviewer for their suggestions. While our current experiments focus on moderate-scale federations to study fundamental collaboration dynamics, we acknowledge the combinatorial complexity challenge. Though computational cost isn't our primary focus given our goal of precise collaborator identification for personalized clients, we recognize scalability limitations commonly in methods like [1]'s loss distance, [2]'s FC-based, or [3]'s gradient-based approaches. This reflects a common challenge in group-wise federated systems where pairwise similarity computations are unavoidable.
>
> To address scalability we plan to use:
>
> 1. **Sinkhorn Approximation**: For large client counts, we employ Sinkhorn's entropy-regularized optimal transport [4] to efficiently approximate similarity matrices through iterative refinement.
>
> 2. **Localized Assessment**: Clients compute similarities only with Top-K neighbors [5] or historical collaborators, using compressed parameter representations/PEFT adapters for efficiency.
>
> 3. **Sparse Matrix Construction**: We retain only high-similarity edges (above threshold) in the similarity graph.
>
> 4. **Algorithmic Optimization**: Our merge-blocking algorithm processes large coalitions first and only traverses non-zero matrix entries, with BC-triggered merging/splitting per Algorithm 1.
>
> In summery, we will leave these promising extensions to future work and clarify this in revision.
>
> [1] M. Zhang, K. Sapra, S. Fidler, S. Yeung, and J. M. Alvarez, "Personalized Federated Learning with First Order Model Optimization," in Proc. Int. Conf. Learn. Represent..
>
> [2] Wu, S., Jia, Y., Liu, B., Xiang, H., Xu, X., & Dou, W. (2025). PFedCS: A Personalized Federated Learning Method for Enhancing Collaboration among Similar Classifiers. .
>
> [3] Sattler, F., Müller, K.-R., and Samek, W. Clustered Federated Learning: Model-Agnostic Distributed Multitask Optimization Under Privacy Constraints. .
>
> [4] Sinkhorn, R. A relationship between arbitrary positive matrices and doubly stochastic matrices. The Annals of Mathematical Statistics, 35(2), 876-879, 1964.
>
> [5] S. Li, T. Zhou, X. Tian and D. Tao, "Learning to Collaborate in Decentralized Learning of Personalized Models," 2022 IEEE/CVF Conference on Computer Vision and Pattern Recognition .
>
> ---
>
> #### ***Weakness 2& Question 2:
>
> **Answer:** Thank you for pointing that which inspires our consideration. 1.Nonlinear Separability: When clients share high feature similarity but opposing labels, the proxy may misgroup them. While rare in practice, XOR-style distributions demonstrate this limitation. 2.Extreme Drift: Abrupt task shifts (e.g., CIFAR-10→MNIST vs EMNIST→CT-RATE) can make knowledge transfer harmful, favoring local training over similarity-based aggregation.
>
>
> ---
>
> #### ***Weakness 3& Question 3***:
>
> **Answer:** Thank you for your sincere remind. We analyze the convergence properties of our method from two perspectives:
>
> 1. **Optimization Convergence at Client Level**:
> problem setting
> \begin{equation}
> \min_{\theta_k}L_k^t(\theta_k)=L_{\mathrm{class}}^t(\theta_k)+\lambda L_{\mathrm{dis}}^t(\theta_k)
> \end{equation}
> client update:
> \begin{equation}
> \theta_{k}^{\tau+1}=\sum_{i\in S}\alpha_{i}(\theta_{i}^{\tau}+\Delta\theta_{i}^{\tau})
> \end{equation}
> assumptions:
> 1.L-smoothness
> \begin{equation}
> L_k^t(y) \leq L_k^t(x) + \langle \nabla L_k^t(x), y - x \rangle + \frac{L}{2} \|y - x\|^2
> \end{equation}
>
> 2.bounded variance
>
> \begin{equation}
> \mathbb{E}_\xi \left[\|\nabla L_k^t(x, \xi) - \nabla L_k^t(x)\|^2\right] \leq \sigma^2
> \end{equation}
>
> 3.unbiased estimation
>
> \begin{equation}
> \mathbb{E}_\xi \left[\nabla L_k^t(x, \xi)\right] = \nabla L_k^t(x)
> \end{equation}
>
> proof:
>
> \begin{align}
> \theta_k^{{\tau}+1} &= \sum_{i\in S} \alpha_i (\theta_i^{\tau} + \Delta\theta_i^{\tau}), \quad \Delta\theta_i^{\tau} = -\eta \nabla_\theta L_k^{\tau} (D_k^{\tau}, \theta_k^{\tau}) \\\\
> &= \alpha_k (\theta_k^\tau + \Delta\theta_k^\tau) + \sum_{i\in S} \alpha_i (\theta_i^\tau + \Delta\theta_i^\tau), \quad \sum_{i\in S} \alpha_i = 1 \\\\
> &= \theta_k^\tau + \Delta\theta_k^\tau - \sum_{i\in S \setminus \{k\}} \alpha_i (\theta_k^\tau + \Delta\theta_k^\tau) + \sum_{i\in S \setminus \{k\}} \alpha_i (\theta_i^\tau + \Delta\theta_i^\tau) \\\\
> &= \theta_k^\tau + \Delta\theta_k^\tau + \sum_{i\in S \setminus \{k\}} \alpha_i [\theta_i^\tau - \theta_k^\tau + \Delta\theta_i^\tau - \Delta\theta_k^\tau] \\\\
> &= \theta_k^\tau - \eta \nabla_\theta L_k^\tau (D_k^\tau, \theta_k^\tau) \\\\
> & + \sum_{i\in S \setminus \{k\}} \alpha_i [\theta_i^\tau - \theta_k^\tau - \eta (\nabla_\theta L_i^\tau (D_i^\tau, \theta_i^\tau) - \nabla_\theta L_k^\tau (D_k^\tau, \theta_k^\tau))]
> \end{align}
> define the disturbance term:
> \begin{equation}
> \delta_k^\tau = \sum_{i\in S \setminus \{k\}} \alpha_i [\theta_i^\tau - \theta_k^\tau - \eta (\nabla_\theta L_i^\tau (D_i^\tau, \theta_i^\tau)) - \nabla_\theta L_k^\tau (D_k^\tau, \theta_k^\tau)]
> \end{equation}
> norm upper bound of disturbance term:
> \begin{equation}
> \| \delta_k^\tau \| \leq \epsilon := \sup \| \delta_k^\tau \|
> \end{equation}
> where
> \begin{align}
> \epsilon_{\text{model}} &= \left\| \sum_{i\in S \setminus \{k\}} \alpha_i (\theta_i^\tau - \theta_k^\tau) \right\| \\\\
> \epsilon_{\text{grad}} &= \eta \left\| \sum_{i\in S \setminus \{k\}} \alpha_i \left(\nabla_\theta L_i^\tau (D_i^\tau, \theta_i^\tau) - \nabla_\theta L_k^\tau (D_k^\tau, \theta_k^\tau)\right) \right\| \\\\
> \text{satisfying} & \quad \| \delta_k^\tau \| \leq \epsilon_{\text{model}} + \epsilon_{\text{grad}} = \epsilon
> \end{align}
> In our dynamic cooperation, $S$ differs with times.
> \begin{align}
> \theta_k^{\tau+1} - \theta_k^\tau &= -\eta \nabla L_k^\tau(\theta_k^\tau) + \delta_k^\tau \\\\
> L_k(\theta_k^{\tau+1}) &\leq L_k(\theta_k^\tau) + \langle \nabla L_k(\theta_k^\tau), -\eta \nabla L_k(\theta_k^\tau; \zeta_k^\tau) + \delta_k^\tau \rangle \\\\
> &\quad + \frac{L}{2} \| -\eta \nabla L_k(\theta_k^\tau, \zeta_k^\tau) + \delta_k^\tau \|^2 \\\\
> \mathbb{E}[L_k(\theta_k^{\tau+1})] &\leq \mathbb{E}[L_k(\theta_k^\tau)] - \eta \mathbb{E}[\|\nabla L_k(\theta_k^\tau)\|^2] + \mathbb{E}[\langle \nabla L_k(\theta_k^\tau), \delta_k^\tau \rangle] \\\\
> &\quad + \frac{L\eta^2}{2} \mathbb{E}[\|\nabla L_k(\theta_k^\tau, \zeta_k^\tau)\|^2] + \frac{L}{2} \mathbb{E}[\|\delta_k^\tau\|^2] \\
> &\quad + L\eta \mathbb{E}[\langle \nabla L_k(\theta_k^\tau, \zeta_k^\tau), \delta_k^\tau \rangle]
> \end{align}
> with bounded conditions:
> \begin{align}
> \mathbb{E}[\|\nabla L_k(\theta_k^\tau, \zeta_k^\tau)\|^2] &\leq \|\nabla L_k(\theta_k^\tau)\|^2 + \sigma^2 \\\\
> \|\delta_k^\tau\|^2 &\leq \epsilon^2
> \end{align}
> we have:
> \begin{align}
> \mathbb{E}[L_k(\theta_k^{\tau+1})] &\leq \mathbb{E}[L_k(\theta_k^\tau)] \\\\
> &- (\eta - C_1\eta^2 - C_2\eta \epsilon - C_3 \epsilon) \mathbb{E}[\|\nabla L_k(\theta_k^\tau)\|^2] \\\\
> &+ C_4 (\eta^2\sigma^2 + \epsilon^2)
> \end{align}
> set:
> \begin{align}
> \eta &= \frac{1}{\sqrt{\tau}} \quad \Delta := L_k(\theta_k^0) - L_k^* \leq C \\\\
> \gamma &:= \eta - C_1\eta^2 - C_2\eta \epsilon - C_3 \epsilon \\\\
> \tau &= 0 \cdots T-1 \\\\
> \mathbb{E}[L_k(\theta_k^{\tau})] &\leq \mathbb{E}[L_k(\theta_k^0)] - \sum_{\tau=0}^{T-1} \gamma \mathbb{E}[\|\nabla L_k(\theta_k^\tau)\|^2] \\\\
> &+ TC_4 (\eta^2\sigma^2 + \epsilon^2) \\\\
> \sum_{\tau=0}^{T-1} \mathbb{E}[\|\nabla L_k(\theta_k^\tau)\|^2] &\leq \frac{L_k(\theta_k^0) - L_k(\theta_k^{\tau})}{\gamma} + \frac{TC_4}{\gamma} (\eta^2\sigma^2 + \epsilon^2) \\\\
> \frac{1}{T} \sum_{\tau=0}^{T-1} \mathbb{E}[\|\nabla L_k(\theta_k^\tau)\|^2] &\leq O\left(\frac{1+\sigma^{2}+\epsilon^{2}}{\sqrt{T}}\right)
> \end{align}
> which indicates optimization convergence on each client.
>
> 2. **Coalition evolution convergence**: Our merge-blocking strategy ensures that the coalition structure evolves toward a stable state.
>
> ---
>
> #### ***Weakness 4& Question 4:
>
> **Answer:** Our framework design explicitly addresses this concern through experimental validation of coalition update frequencies. Results across 10-communication-round task phases demonstrate that per-round coalition evaluation, while computationally intensive, is essential for handling unpredictable data arrival patterns during task learning.
>
> | Update Times | 1 | 1 | 2|2|  5| 5| 10|10|
> |--------------|---------------|---------------|---------------|----------------|---------------|---------------|---------------|----------------|
> || Accuracy   | Forget | Accuracy  | Forget | Accuracy   | Forget | Accuracy    | Forget |
> | EMNIST-LTP         | 42.4 | 15.6   | 49.3 | 13.1   | 50.6 | 10.4   | **52.5**  | **8.9**    |
> | EMNIST-shuffle   | 70.9 | 12.9   | 71.5 | 12.3   | 74.3 | 9.6    | **78.3**  | **4.2**    |
>
> Timely collaborator identification is critical as mini-batches arrive during task training. Infrequent coalition updates risk failing to adapt to emerging patterns - e.g., when learning new classes like "dog", delayed updates maintain outdated collaborations, increasing catastrophic forgetting and accuracy degradation.
>
> ---
>
> #### ***Limitations 2:***
> **Answer:** Thank you. We acknowledge this parameter sensitivity and plan to implement auto-tuning in future work.
>
> ---
>
> #### ***Limitations 3***
>
> **Answer:** Thank you. In practice, an impartial third-party (e.g., industry association) handles model collection, benefit assessment, and cooperation publishing in our federated learning paradigm. Crucially, our approach differs from centralized learning in two aspects: Only evaluates optimal coalitions (preventing self-interest) without performing model aggregation/distribution, which remains client-autonomous. Preserves client-specific learning trajectories via decentralized knowledge transfer, avoiding global model imposition.
>
> We appreciate the privacy concerns raised and will address trusted third-party aspects in future work.

---

### Official Review · Reviewer_88x1 · 2025-07-02

**Clarity:** 3
**Significance:** 3
**Originality:** 3
**Rating:** 4
**Confidence:** 4

**Summary:**

This paper presents a novel decentralized dynamic cooperation framework (DCFCL) to address the challenges of temporal and spatial catastrophic forgetting in Federated Continual Learning (FCL). The authors formulate client cooperation using coalitional game theory, where each client dynamically joins coalitions based on estimated mutual benefit, evaluated via an overall similarity metric combining gradient coherence and model similarity. A merge-blocking algorithm and dynamic cooperative evolution strategy are proposed to ensure convergence to a cooperative equilibrium in each aggregation round. Extensive experiments across several datasets demonstrate the superiority of the proposed method over existing FL, CL, and FCL baselines.

**Questions:**

as above

**Ethical Concerns:**

["NO or VERY MINOR ethics concerns only"]

**Final Justification:**

The authors have further elaborated the concepts, time complexity, space complexity, and hyperparameter robustness in the paper, and have basically answered the review comments. Considering the overall contribution of the paper, I maintain the  previous score.

**Limitations:**

yes

**Quality:**

3

**Strengths And Weaknesses:**

Strength
1. The paper introduces a novel application of cooperative game theory in the FCL context, enabling clients to dynamically form non-overlapping coalitions. This design effectively balances knowledge retention and acquisition under data heterogeneity.
2. The proposed overall similarity metric, which combines gradient and model-level coherence, is both theoretically grounded and computationally efficient. The merge-blocking and dynamic evolution algorithms are well-designed to support convergence to cooperative equilibrium.
3. The method is rigorously evaluated on four benchmark datasets, including highly heterogeneous and multimodal settings. The authors also conduct thorough ablation studies and parameter sensitivity analyses to demonstrate the individual contributions of key components.

Weakness
1. In the Introduction, the authors claim that spatial catastrophic forgetting is caused by aggregation. However, the subsequent preliminary experiments demonstrate that federated aggregation helps alleviate forgetting, which appears contradictory. Could the authors clarify this seeming inconsistency?

2. The manuscript does not provide a clear definition of Personalized Federated Continual Learning (FCL). What exactly distinguishes personalized FCL from decentralized FCL? A clearer conceptual distinction between the two is necessary.

3. Regarding the client grouping strategy for utility computation: is it based on exhaustive enumeration or another approach? Given that the benefit comparison involves dynamic equilibrium, which can be computationally expensive, are there any strategies proposed to reduce time complexity?

4. Does the proposed method involve storing state spaces? If so, what is the associated space complexity?

5. In Table 3, two important hyperparameters significantly affect model performance. The selection of these hyperparameters is critical for practical deployment. Additionally, on the EMNIST-shuffle dataset, model performance appears more sensitive to these hyperparameters. Is there any analysis or discussion provided on this issue?

---

> ### Author Rebuttal · Authors · 2025-07-31
>
> ## Reviewer 88x1
>
> We would like to thank the reviewer for the useful comments. Below are our responses to the comments in **Weaknesses**.
>
> ---
>
> #### ***Weakness 1: In the Introduction, the authors claim that spatial catastrophic forgetting is caused by aggregation. However, the subsequent preliminary experiments demonstrate that federated aggregation helps alleviate forgetting, which appears contradictory. Could the authors clarify this seeming inconsistency?***
>
> ****Answer:**** Thanks for the useful comments. We will clarify our statement from following aspects.
>
> 1. **Federated Aggregation Benefit**: We acknowledge that federated aggregation can mitigate forgetting, as knowledge from other clients helps recall prior tasks. This is illustrated in Section 2, Paragraph 2, via Fig. 1(a)(b), highlighting FCL’s value and motivating the use of central aggregation [1,2].
>
> 2. **Critical Limitation**: However, we question whether aggregating all models is always beneficial. We argue that blind aggregation is suboptimal—especially under strong spatial heterogeneity, which may cause severe knowledge interference, termed spatial catastrophic forgetting in prior work [3,4].
>
> 3. **Empirical Validation and Motivation**: As shown in Paragraph 3 and Fig. 1\(c\), we empirically verify this concern. We emphasize that selective and cooperative aggregation is essential for personalized continual learning, motivating the design of our DCFCL framework.
>
> [1] Yoon, J., Jeong, W., Lee, G., Yang, E., & Hwang, S. J. (2021). Federated continual learning with weighted inter-client transfer. In Proceedings of the 38th International Conference on Machine Learning (pp. 12073-12086). PMLR.
>
> [2] Shenaj, D., Toldo, M., Rigon, A., & Zanuttigh, P. (2023). Asynchronous federated continual learning. In *2023 IEEE/CVF Conference on Computer Vision and Pattern Recognition Workshops. IEEE.
>
> [3] X. Yang, H. Yu, X. Gao, H. Wang, J. Zhang and T. Li, "Federated Continual Learning via Knowledge Fusion: A Survey," in IEEE Transactions on Knowledge and Data Engineering, vol. 36, no. 8, pp. 3832-3850, Aug. 2024.
>
> ---
>
> #### ***Weakness 2: The manuscript does not provide a clear definition of Personalized Federated Continual Learning (FCL). What exactly distinguishes personalized FCL from decentralized FCL? A clearer conceptual distinction between the two is necessary.***
>
> ****Answer:**** Thanks for the useful comments. Here we give the definition of two concepts.
>
> 1.**Personalized FCL**: Personalized FCL is a scenario where clients aim to preserve their individual task histories and learn models tailored to their own data distributions across time. In PFCL, clients can share models to facilitate their own performance [1,2].
>
> 2.**Decentralized Learning**: Decentralized learning refers to the absence of a central server in communication topology [2,3]. Clients spontaneously aggregate models and don't rely on central server. Our work emphasizes the personalized continual learning objective under decentralized settings.
>
> 3.**Framework Objectives and Rationale**: Since our personalized FCL framework is based on dynamic cooperation, the objective is to optimize the performance of each rational client rather than a global objective as shown in Eq.1.:
> \begin{equation}
> \underset{\theta_k^t}{\operatorname*{\operatorname*{argmin}}}\underset{\mathcal{D}_k^t}{\operatorname*{\operatorname*{E}}}[L_k^t(\mathcal{D}_k^t,\theta_k^t)]
> \end{equation}
> which naturally requires no central architecture. Thus, we argue decentralized topology better suits personalized FCL than centralized, as discussed in Section 1 (Paragraphs 2-3).
>
> 4. **Revision Plans**: In the revised version, we will make two definitions more precise in Section 1.
>
> [1] J. Xie, C. Zhu, and S. Li, "FedMeS: Personalized Federated Continual Learning Leveraging Local Memory," arXiv preprint arXiv:2404.12710, 2024.
> [2] S. Li, T. Zhou, X. Tian and D. Tao, "Learning to Collaborate in Decentralized Learning of Personalized Models," 2022 IEEE/CVF Conference on Computer Vision and Pattern Recognition, New Orleans, 2022.
>
> ---
>
> #### ***Weakness 3: Regarding the client grouping strategy for utility computation: is it based on exhaustive enumeration or another approach? Given that the benefit comparison involves dynamic equilibrium, which can be computationally expensive, are there any strategies proposed to reduce time complexity?***
>
> ****Answer:**** Thanks for the comments. Our coalition formation avoids exhaustive enumeration. Instead of exploring all grouping states, we iteratively merge coalitions under the **PT condition** (Profitable Transition: merging occurs only when clients gain higher benefits). For complexity reduction, we employ these strategies:
>
> 1. **Benefit Calculation via Coalitional Affinity Game**: To avoid prohibitive computation/communication costs from exhaustive local aggregation, we propose overall similarity to quantify pairwise client benefits. Using coalitional affinity game, we prove multi-client benefits can be derived from 2-client relationships via function $f(\cdot)$ (Appendix A). This reduces complexity from $O(\sum_{k=1}^K C_K^k)$ to $O(C_K^2)$ (Appendix B.4).
>
> 2. **Merge-Blocking Algorithm**: Our iterative algorithm (Section 4.2) avoids traversing the exponential partition space. While each iteration remains $O(K2^K)$, it significantly improves over the $O(B_K 2^K)$ complexity of full state-space traversal.
>
> 3. **Pruning in Merge-Blocking**: Redundant subset comparisons are eliminated each round. Empirical results show fast convergence regardless of client count, with formation times substantially reduced versus non-optimized approaches.
>
>
> | Number of clients | 10| 20| 30| 40| 50 |
> |:-------------------:|:-------:|:-------:|:--------:|:-------:|:-------:|
> | Time (min)          | 0.116  | 2.715  | 8.279  |12.526  | 27.15  |
> | Iterations        | 2      | 2      | 3       | 3  | 5  |
> | Time (min)-w/o MBA         | 1.438  | 8.504  | 24.733  | 39.293  | 54.596  |
>
> ---
>
> #### ***Weakness 4: Does the proposed method involve storing state spaces? If so, what is the associated space complexity?***
>
> ****Answer:**** We thank the reviewer for raising this concern. Our method avoids explicit storage of the entire state space by maintaining only a small cache of previous-round model states and utility values for current candidate coalitions. The space complexity analysis shows this design achieves linear scaling: each client requires $O(d)$ local storage for knowledge distillation (significantly less than rehearsal-based CL approaches that store prior task data/generators), while the third-party coordinator maintains $O(2Kd)$ space for current/previous models of all clients (detailed in Section 4.3 and Appendix B.1). We will further clarify this space complexity analysis in the final manuscript.
>
>
> ---
>
> #### ***Weakness 5: In Table 3, two important hyperparameters significantly affect model performance. The selection of these hyperparameters is critical for practical deployment. Additionally, on the EMNIST-shuffle dataset, model performance appears more sensitive to these hyperparameters. Is there any analysis or discussion provided on this issue?***
>
> ****Answer:**** We thank the reviewer for raising this concern. To further illustrate, we conduct an extended sensitivity analysis in the following tables.
>
> **Sensitivity Analysis of λ and ε on MNIST-SVHN-F**
>
> | λ\ε | 0.0   | 0.2   | 0.4   | 0.6   | 0.8   | 1.0   |
> |:---:|:-----:|:-----:|:-----:|:-----:|:-----:|:-----:|
> | **0.0** | 60.5 | 62.1 | 63.8 | 63.1 | 61.7 | 60.9 |
> | **0.2** | 61.2 | 61.4 | 64.9 | 62.4 | 62.5 | 61.3 |
> | **0.4** | 62.0 | 64.2 | 65.7 | 64.7 | 63.2 | 62.1 |
> | **0.6** | 60.1 | 64.8 | 66.1 | 64.0 | 63.9 | 62.8 |
> | **0.8** | 63.8 | 64.3 | 66.5 | 65.2 | 64.3 | 61.2 |
> | **1.0** | 64.5 | 65.9 | **66.7** | 65.8 | 64.9 | 62.5 |
>
> We compared the maximum fluctuation in accuracy when one value was fixed and the other changed.
> $\lambda=0.4,max(Range_\varepsilon) =3.6$
> $\varepsilon =0.2,max(Range_\lambda) =3.8$
>
> **Sensitivity Analysis of λ and ε on EMNIST-Shuffle**
> | λ\ε | 0.0   | 0.2   | 0.4   | 0.6   | 0.8   | 1.0   |
> |:---------:|:---------:|:---------:|:---------:|:---------:|:---------:|:---------:|
> | **0.0** | 70.2 | 73.2 | 81.9 | 73.6 | 74.1 | 64.2 |
> | **0.2** | 75.9 | 78.3 | 80.2 | 70.4 | 71.4 | 72.2 |
> | **0.4** | 78.7 | 78.9 | 82.9 | 71.2 | 70.9 | 70.6 |
> | **0.6** | 78.0 | 79.6 | 82.7 | 73.7 | 71.1 | 71.0 |
> | **0.8** | 75.2 | 79.7 | 81.7 | 74.2 | 73.2 | 74.2 |
> | **1.0** | 73.1 | 81.3 | **84.5** | 80.7 | 76.9 | 74.5 |
>
> Similarly, for the emnist-shuffle dataset:
> $\lambda =0.0,max(Range_\varepsilon) =17.7$
> $\varepsilon =1.0,max(Range_\lambda) =10.3$
>
> EMNIST-shuffle shows greater sensitivity to $\lambda$ and $\varepsilon$ due to extreme class imbalance, requiring larger $\varepsilon$ to better leverage global model information for collaborator identification - highlighting the critical importance of proper similarity weight settings for highly heterogeneous datasets.

---

> > ### Comment · Reviewer_88x1 · 2025-08-05
> >
> > The authors have adequately addressed all my previous comments and concerns. I have no further suggestions.  I maintain my original score.

---

> > > ### Author Response · Authors · 2025-08-06
> > >
> > > **Thanks for Reviewer's Feedback**
> > >
> > > Dear Reviewer,
> > >
> > > Thank you for your time and valuable comments on our manuscript. We sincerely appreciate your acknowledgment of our paper and your continued support. Your constructive feedback has greatly improved our work.
> > >
> > > Best regards,
> > >
> > > Authors

---

### Official Review · Reviewer_u9Pu · 2025-07-07

**Clarity:** 3
**Significance:** 3
**Originality:** 3
**Rating:** 4
**Confidence:** 4

**Summary:**

This paper proposes a federated continual learning method that estimates client benefits in cooperative coalitions by estimating task similarity and using a coalitional affinity game. In addition, this paper designs an algorithm to efficiently find and update the cooperative equilibrium of clients.

**Questions:**

1. Why is KL loss needed to achieve a better estimation of the coordination ability between tasks? In order to learn the task, the model deviates more from the previous round model, which may affect the aggregation effect, but this article does not explain how it is related to identifying cooperators.


2. In the fields of FL, CL, and FCL, there are many methods for estimating task similarity and selecting clusters. How to explain that the method proposed in this article is better than them?

**Ethical Concerns:**

["NO or VERY MINOR ethics concerns only"]

**Limitations:**

Yes.

**Paper Formatting Concerns:**

None.

**Quality:**

3

**Strengths And Weaknesses:**

Strengths:
1. The motivation of this paper is clear, and an aggregation algorithm with certain theoretical guarantees is proposed to select the scope of aggregation for each task, effectively avoiding the knowledge conflict between tasks.

2. This paper conducts relatively sufficient experiments on multiple data sets, which can illustrate the effectiveness of the proposed algorithm in resolving conflicts between tasks.

Weaknesses:
1. In the related work section, the reviewer believes that there are currently a large number of methods based on pre-trained models and parameter-efficient fine-tuning in CL and FCL. Although this article focuses on the effect of learning from scratch, it is necessary to discuss such methods in related work and explain the advantages of learning from scratch.

2. In the field of continual learning, forgetting is an indicator often used to measure the ability of an algorithm to avoid catastrophic forgetting. Because the learning capabilities of different algorithms may vary, the reviewer believes that it is necessary to provide a forgetting indicator.

---

> ### Author Rebuttal · Authors · 2025-07-31
>
> **Response to W1**:
>
> ****Answer:**** We thank the reviewer for their insightful suggestion. While our current work investigates continual learning from scratch to analyze task coordination in a controlled setting, we acknowledge the growing importance of pre-trained models (e.g., LLMs, vision backbones) [6] and parameter-efficient tuning (e.g., adapters, LoRA) [7] in practical FCL. Here we concisely address this concern from three perspectives:
> 1. **Method Comparision**: While these methods benefit from strong initialization and existing knowledge, leading to reduced training costs and improved early-stage performance, our approach concentrates on learning from scratch. This allows us to directly study coalition-based knowledge retention and generalization dynamics in federated continual learning without relying on pre-trained priors. Notably, our method is complementary to PEFT-based approaches and could potentially be integrated with them in future work.
> 2. **Revised Version**: In the revised version, we will add a dedicated paragraph in the Related Work section discussing these approaches.
> 3. **Method Extension**: To demonstrate our framework's adaptability to pre-trained foundation models, we conducted preliminary experiments applying our DCFCL approach with PEFT. Specifically:Replaced local models with a pre-trained ViT-B/16 backbone (iBOT-21K). Implemented adapter-based PEFT, where only PEFT parameters are updated and transmitted. Maintained our original coalition similarity computation using PEFT parameter embeddings. Preserved our benefit calculation and coalition evolution framework. Compared against FedAvg+adapter [6] and ClusterFL+adapter [8] baselines on CIFAR100. Our experimental results are as follows
>
> |Method|Accuracy (↑)|Forget (↓)|
> |-|-|-|
> |FedAvg+adapter|69.8|6.9|
> |Cluster+adapter|73.1|7.2|
> |DCFCL+adapter|77.5|4.2|
>
> As shown in the Table, **DCFCL+Adapter** achieves the best accuracy and the lowest forgetting rate on CIFAR-100. These results validate the generalizability of our mechanism in PEFT scenarios.
>
> ---
>
> **Response to W2**:
>
> ****Answer****: Thank you for pointing this out. In our experiments, we follow recent works [9,10] and use average accuracy and average forgetting as evaluation metrics. Let $\alpha_k^{t,i}$ denote the test accuracy on task $i$ after learning task $t$ in client $k$. Average forgetting measures backward transfer, defined as the difference between peak and final accuracy of each task. A weighted average is applied in its calculation:
> \begin{equation}
> \text{Average Forgetting} = \frac{1}{\sum_{k=1}^{K}\sum_{i=1}^{T-1} n_k^i} \sum_{k=1}^{K}\sum_{i=1}^{T-1} \max_{t \in \{1,\cdots,T-1\}} (a_k^{t,i} - a_k^{T,i}){n_k^i}
> \end{equation}
> In our revision, we will supplement this point in the experimental section.
>
> ---
> **Response to Q1**
>
> ****Answer:****  Thank you for your valuable comment. We would like to clarify a possible misunderstanding regarding the use of **KL loss** in our method.
> 1. **Clarification**: Our method does not rely on KL divergence to measure task coordination. Instead, knowledge distillation is used as an auxiliary tool to preserve feature consistency during continual learning (Section 4.1). As new tasks alter the classifier, it becomes harder to identify cooperators for knowledge recall. By maintaining feature space consistency, we enable efficient cooperator matching using existing model information, with minimal communication overhead.
> 2. **Empirical Validation**: We experimentally verify that our method, with KD as an auxiliary tool, better mitigates catastrophic forgetting than naive KD. Even applying KD to all clients fails to achieve optimal performance without our framework, as shown in Section C.2, line 530 (*"To control variables during local training, we adopted the same KD method into all FL baselines, which still obtained suboptimal results."*).
> 2. **Relation to Identifying Cooperators**: We use knowledge distillation between the classifier’s current and past outputs to maintain feature space consistency and prevent drift during continual learning. This helps retain discriminative features from previous tasks, enabling accurate identification of similar clients without additional model exchange. As shown in **Fig.2**, clients choose different cooperators at each task stage based on the **trade-off** between acquiring new knowledge and retaining prior learning. KD helps clients find suitable partners to preserve prior knowledge. In short, distillation is not for coordination estimation but for **preserving useful representations** to support **similarity-based** cooperation under dynamic, heterogeneous conditions.
>
> ---
> **Respond to Q2**
>
> ****Answer:****  Thank you for raising this. Based on our investigation, many existing works estimate task similarity via the following several mainstream metrics:
>
> 1. **L2-based**: the $l_2$ distance across client models. [1]
> 2. **Gradient-based**: only use the cosine similarity of client gradients.[2,3]
> 3. **EDC**: truncate gradients onto key directions via Singular Value Decomposition and computes similarity in this reduced space. [4]
> 4. **FC-based**: the cosine similarity between client classifier parameters.[5]
>
> Under our experimental settings, we only replace the **overall similarity** part. The experimental results are as follows:
>
> |Method| EMNIST-LTP| EMNIST-shuffle|CIFAR100| MNIST-SVHN-F|
> |-|-|-|-|-|
> |  |Accuracy|Forget|Accuracy|Forget|Accuracy|Forget|Accuracy|Forget|
> |Grad-based|45.3|14.8|73.7|15.7|32.6|10.3|55.5|17.1|
> |EDC|49.8|10.4|72.9|13.6|32.9|8.2|60.4|12.2|
> |L2-based|50.5|12.4|73.2|12.0|33.7|8.4|65.8|14.2|
> |FC-based|51.4|**8.8**|77.5|7.6|37.5|6.8|62.5|10.5|
> |DCFCL|**52.5**|8.9|**78.3**|**4.2**|**40.4**|**4.7**|**66.7**|**6.9**|
>
> Using gradient-based methods alone leads to severe forgetting, as gradient similarity struggles to identify models aligned with prior tasks, hindering knowledge retention. EDC reduces computation by truncating the gradient matrix and approximating similarity but yields suboptimal performance. In contrast, our overall similarity accurately identifies cooperators and evaluates benefits, achieving the best results in both accuracy and forget rate.
>
> [1] M. Xie, G. Long, T. Shen, T. Zhou, X. Wang, and J. Jiang, “Multi-center federated learning,” arXiv preprint arXiv:2005.01026, 2020.
>
> [2]  Sattler, F., Müller, K.-R., and Samek, W. Clustered Federated Learning: Model-Agnostic Distributed Multitask Optimization Under Privacy Constraints. IEEE Transactions on Neural Networks and Learning Systems, 32(8): 3710-3722, 2021. doi: 10.1109/TNNLS.2020.3015958.
>
> [3] Li, Z., Lin, T., Shang, X., & Wu, C. (2023). Revisiting Weighted Aggregation in Federated Learning with Neural Networks. arXiv preprint arXiv:2302.10911.
>
> [4] M. Duan, D. Liu, X. Ji, R. Liu, L. Liang, X. Chen, and Y. Tan, "FedGroup: Efficient Federated Learning via Decomposed Similarity-Based Clustering," in Proc. IEEE Int. Symp. Parallel Distrib. Process. Appl , 2021, pp. 228-237.
>
> [5] Wu, S., Jia, Y., Liu, B., Xiang, H., Xu, X., & Dou, W. (2025). PFedCS: A Personalized Federated Learning Method for Enhancing Collaboration among Similar Classifiers. Proceedings of the AAAI Conference on Artificial Intelligence, 39(20), 21572-21580.
>
> [6] Cai, D., Wu, Y., Wang, S., Lin, F. X., and Xu, M. FedAdapter: Efficient Federated Learning for Modern NLP. arXiv preprint arXiv:2205.10162, 2023.
>
> [7] Yi, L., Yu, H., Wang, G., Liu, X., and Li, X. pFedLoRA: Model-Heterogeneous Personalized Federated Learning with LoRA Tuning. arXiv preprint arXiv:2310.13283, 2024.
>
> [8] Sattler, F., Müller, K.-R., and Samek, W. Clustered Federated Learning: Model-Agnostic Distributed Multitask Optimization Under Privacy Constraints. IEEE Transactions on Neural Networks and Learning Systems, 32(8): 3710-3722, 2021. doi: 10.1109/TNNLS.2020.3015958.
>
> [9] Wuerkaixi, A., Cui, S., Zhang, J., Yan, K., Han, B., Niu, G., Fang, L., Zhang, C., and Sugiyama, M. Accurate Forgetting for Heterogeneous Federated Continual Learning. In The Twelfth International Conference on Learning Representations (ICLR), 2024.
>
> [10] Qi, D., Zhao, H., and Li, S. Better Generative Replay for Continual Federated Learning. In The Eleventh International Conference on Learning Representations (ICLR), 2023.

---

### Comment · Area_Chair_HEAf · 2025-08-04
**Gentle reminder**

Dear reviewers,

If you haven't done yet, please review the authors' rebuttal and reach out to them if you have any additional concerns. The reviewer-author discussion will end on Aug 6 11:59pm AoE.

Best Regards,

AC

---

> ### Comment · Area_Chair_HEAf · 2025-08-07
> **Author-reviewer discussion ending on Aug. 8th**
>
> Dear reviewers,
>
> If you have discussed well with the authors regarding the potential concerns, thank you very much for your effort.
>
> If you haven't done so, please follow the official review policy of NeurIPS 2025 during the author-review discussions.
>
> - **Reviewers must participate in discussions with authors before submitting “Mandatory Acknowledgement”**. The non-participating reviewers will receive possible penalties of this year's responsible reviewing initiative and future reviewing invitations.
> - Clicking “Mandatory Acknowledgement” prematurely does NOT release Reviewers from participating in discussions.
> - Please note “Mandatory Acknowledgement” button is to be submitted only when reviewers fulfill all conditions below (conditions in the acknowledgment form):
>     - read the author rebuttal
>     - engage in discussions (reviewers must talk to authors, and optionally to other reviewers and AC - ask questions, listen to answers, and respond to authors)
>     - fill in "Final Justification" text box and update “Rating” accordingly (this can be done upon convergence - reviewer must communicate with authors first)
> - To facilitate discussions, the deadline of Author-Reviewer discussions has been extended to **Aug 8, 11.59pm AoE**. Please discuss with the authors regarding your concerns about this submission.
>
> Best,
>
> AC

---

### Note · Authors · 2025-08-15

We thank the reviewers for their constructive feedback and for raising their scores. We are pleased that the following strengths are recognized: novelty (reviewer 88x1, ckij), clear motivation (u9Pu, 88x1, ckij), sufficient experiments (u9Pu, 88x1, ckij), significance (ckij).

During discussion, most concerns were addressed:

We thank Reviewer u9Pu for raising:

1.PEFT method: we extended experiments with adapter-based PEFT, showing scalability.

2.Similarity metrics: we evaluated 4 mainstream similarity metrics, demonstrating our method’s advancement.

3.Forgetting metric: we will supplement the definition in revision.

We thank Reviewer 88x1 for raising:

1.Parameter sensitivity: we extended fine-grained experiments, showing sensitivity changes under different heterogeneities.

2.Complexity reduction: we clarified avoidance of enumeration, restated time/space complexity analyses.

3.Concept clarification: we further explained PFL, decentralization, and forgetting-aggregation relationship and will add a notion interpretation for PFL in revision.

We thank Reviewer ckij for raising:

1.Complexity: Our method's core objective of finding optimal client cooperation in federated learning inherently involves computational complexity, as the problem is NP-hard by nature. While delivering optimal performance at smaller scales, large-scale applications require approximate solutions that balance performance with computational feasibility. To address this, we introduced approximation strategies.

2.Convergence: we added detailed convergence analysis.

3.Coalition update frequency: we showed an experiment and demonstrated per-round update is essential.

We thank Reviewer 1yUZ for raising:

1.Concept concerns: we expanded explanations on PFL and decentralized topologies.  We clarified Section 1's logical flow connecting to  identified issues and our proposed solutions, while distinguishing our original methodological contributions from prior works.

2.Objective: we explained rationality of eq.1 and $u_k$ via decomposition and will incorporate this process in revision.

3.Complexity: our game-theoretic optimization for rational clients entails more complexity than clustering, yet offers three key advantages: provably optimal solutions, dynamic grouping capability, lower complexity than comparable algorithms. We then offered approximation schemes.

We sincerely thank reviewers for their valuable suggestions and the AC for efforts during rebuttal.

Best wishes,

Authors

---

### Decision · Program_Chairs · 2025-09-17

**Decision:**

Accept (poster)

**Comment:**

This paper introduced a novel decentralized federated continual learning framework. This framework enforced the dynamic cooperation among clients to mitigate catastrophic forgetting and improve model performance.

Strengths:
- The paper is well-motivated.
- The proposed similarity metric for FCL is both theoretically grounded and computationally efficient
- The method is well validated across various benchmarks, followed by detailed ablation studies and parameter sensitivity analyses.

Weakness:
- The conceptual distinction between decentralized FCL and personalized FCL can be further clarified to highlight the FCL problem studied in this work.
- The space complexity analysis can be clarified.
- More insights can be provided to explain the impact of hyperparameters in the experiments.
- The scalability of the proposed method can be analyzed and improved.
- The notation and formulation used in this paper can be further clarified.

Recommendation

Though there are some concerns related to clarity and complexity analysis, all reviewers provided positive feedback overall. I recommend acceptance, and encourage the authors to address the reviewers’ suggestions to further improve the quality and clarity of the paper.